# Advancing Spatiotemporal Representations in Spiking Neural Networks via Parametric Invertible Transformation

**Yinsong Yan**[1], **Yujie Wu**[2], **Jibin Wu**[1,2*]
[1] Department of Data Science and Artificial Intelligence, The Hong Kong Polytechnic University
[2] Department of Computing, The Hong Kong Polytechnic University
`jibin.wu@polyu.edu.hk`

## Abstract

Spiking Neural Networks (SNNs) are regarded as energy-efficient neural architectures due to their event-driven, spike-based computation paradigm. However, existing SNNs suffer from two fundamental limitations: (1) the constrained representational space imposed by binary spike firing mechanisms, which restricts the network's capacity to encode complex spatiotemporal patterns, and (2) the ineffective design of surrogate gradient functions that leads to gradient mismatch issues and suboptimal learning dynamics. To address these challenges, we propose the Parametric Invertible Transformation (PIT), which operates in a conjugate manner with neuronal dynamics to achieve adaptive modulation and augmented spike representations simultaneously. Second, we design an auxiliary gradient correction term to mitigate the gradient mismatch issue and oscillation phenomena during training. Moreover, we introduce a theoretical framework for analyzing the spatiotemporal representation space of SNNs. Extensive experiments on both static and neuromorphic datasets demonstrate state-of-the-art performance with our proposed method. This approach lays the theoretical foundation for expanding the spatiotemporal representations of SNNs, offering a viable pathway for developing low-latency and high-performance neuromorphic processing systems in resource-constrained environments. The code is available at https://github.com/YinsongYan/ICLR26.

## 1 Introduction

Spiking Neural Networks (SNNs) are designed to emulate the neural dynamics of biological systems by utilizing asynchronous spikes for communication (Maass, 1997; Roy et al., 2019). In contrast to Artificial Neural Networks (ANNs), which rely on continuous activations, SNNs operate with binary spikes, providing huge potential for more efficient computations by transforming dense Multiply-And-Accumulate (MAC) operations into sparse Accumulate (AC) operations. Combined with the event-driven computation paradigm, where computations are triggered only upon the receipt of spikes, these characteristics offer inherent sparsity and low energy consumption, making them promising and appealing for scenarios necessitating real-time and energy-constrained processing (Merolla et al., 2014; Davies et al., 2018; Pei et al., 2019; Chen et al., 2023; 2025; Roy et al., 2019).

Despite these advantages, SNNs face two fundamental challenges that limit their practical effectiveness. First, SNNs suffer from constrained representational capacity due to their low-precision binary spike representation. This inherently leads to varying degrees of information loss and performance degradation in complex classification tasks (Deng & Gu, 2021; Guo et al., 2024). To address this issue, prior works have explored advanced spiking neuron models with complex dynamics (Fang et al., 2021b; Yin et al., 2021; 2023; Chen et al., 2024; Hao et al., 2024), normalization techniques (Kim & Panda, 2021; Zheng et al., 2021; Duan et al., 2022), multi-bit spike representations (Guo et al., 2024; Xing et al., 2024; Guo et al., 2022c), attention mechanisms (Yao et al., 2021; 2023b), and modern architectures including ResNet (Fang et al., 2021a; Hu et al., 2024) and spike-driven transformers (Zhou et al., 2023; Yao et al., 2023a; 2024; Zhou et al., 2024; Yao et al., 2025). Nevertheless,

---

*Corresponding author

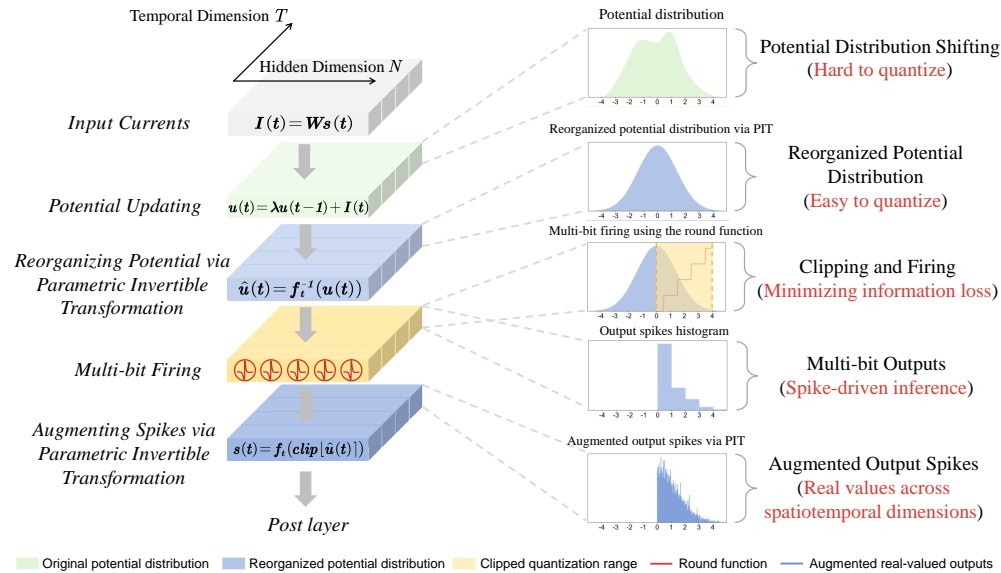

Figure 1: The overall workflow of our proposed method. By incorporating a parametric invertible transformation (PIT) into neuronal dynamics in a conjugate manner, spiking neurons could adaptively reorganize the potential distribution before firing and emit real-valued outputs across spatial and temporal dimensions. It is worth noting that the SNN integrated with PIT could preserve the spike-based inference and event-driven computation paradigm through the reparameterization technique.

these approaches have not adequately addressed the inherent limitations of the representational space of SNNs, particularly when dealing with data streams characterized by high dynamic variations and complex distributions. Second, existing surrogate gradient methods tailored for binary spikes (Wu et al., 2018; Rathi & Roy, 2021; Li et al., 2021) and integer spike neurons (Luo et al., 2024; Yao et al., 2025) exhibit limited adaptability and often introduce gradient mismatch issues and oscillation phenomena during backpropagation (Yin et al., 2019; Spallanzani et al., 2022; Wu et al., 2023b; Nagel et al., 2022).These two limitations collectively underscore the critical need for developing novel approaches that can simultaneously enhance the representational capacity of SNNs while enabling stable and efficient training.

In this study, we focus on enhancing the representational capabilities of SNNs by integrating the parametric invertible transformation (PIT) into neuronal dynamics. As shown in Figure 1, our proposed PIT operates in a conjugate manner before and after the firing operation of spiking neurons, improving spatiotemporal representations while preserving the event-driven and spike-based inference computation paradigm through the reparameterization technique (see Appendix A). To further address the non-differentiable spike firing operation, we propose an auxiliary gradient correction term designed to mitigate the gradient mismatch issue and oscillation phenomena, facilitating rapid convergence and improved generalization. Additionally, we establish a theoretical framework to analyze and measure the spatiotemporal representational capacity of SNNs. Comprehensive experiments on static and neuromorphic visual datasets, including CIFAR10 (Krizhevsky et al., 2009), CIFAR100 (Krizhevsky et al., 2009), DVS-Gesture (Amir et al., 2017), CIFAR10-DVS (Li et al., 2017), and ImageNet-1k (Deng et al., 2009), validate the superiority and effectiveness of our method. Our contributions can be summarized as follows:

- We introduce PIT, a parametric invertible transformation, to enhance the spatiotemporal representations of SNNs in a conjugate manner. PIT employs an input-distribution-aware parameterization strategy to dynamically expand representations while preserving spike-driven inference via the reparameterization technique. Besides, we design a rectified surrogate gradient term for improved gradient flow, enabling fast and stable convergence.

- We provide a theoretical framework to analyze and measure the spatiotemporal representation capacity of SNNs. Based on our theoretical framework, we demonstrate that the representation capacity of SNNs exhibits a logarithmic relationship with both the firing bit numbers and total time steps.

- Extensive experiments on both static and neuromorphic datasets demonstrate the state-of-the-art performance achieved by our method across various architectures. Notably, when incorporating our method into SEW ResNet34, it surpasses the baseline model with the same architecture after training for only one epoch and ultimately improves performance by 5.62% on the ImageNet dataset.

## 2 RELATED WORK

**Training Methods for SNNs.** Current SNN training methods can be broadly categorized into two strategies: (1) ANN-to-SNN conversion, which establishes mathematical mappings between ReLU activation layers and spiking neuron layers, and (2) direct training, which employs the Backpropagation Through Time (BPTT) algorithm. Direct training allows SNNs to operate with an extremely small time window by employing BPTT alongside surrogate gradient (SG) functions to approximate the non-differentiable firing operation (Wu et al., 2018; Neftci et al., 2019; Zenke & Vogels, 2021; Lee et al., 2020). Instabilities during deep SNN training have been mitigated using normalization techniques (Kim & Panda, 2021; Zheng et al., 2021; Duan et al., 2022; Guo et al., 2023b), regularization terms to maximize output entropy (Guo et al., 2022a), membrane potential rearrangement (Guo et al., 2023a; 2022b), and temporal efficient gradient re-weighting (Deng et al., 2022). For integer spike forms of SNNs, prior works (Luo et al., 2024; Yao et al., 2025) directly utilize the clipped rectangular surrogate gradient function to achieve direct training. However, this approach may result in oscillation phenomena that hinder the network from converging to a solution with good generalizability (Yin et al., 2019; Spallanzani et al., 2022; Wu et al., 2023b; Nagel et al., 2022). Nevertheless, most existing methods are tailored to binary spike forms of SNNs, while effective and stable training methods for SNNs with integer spike forms remain underexplored. This gap motivates us to design corresponding parameterization techniques and surrogate gradients to enable effective and stable training, particularly in deep networks.

**Enhancing Spike Representations for SNNs.** At the neuron level, previous works have introduced learnable parameters into spiking neuron models, including learnable membrane decay factors (Bellec et al., 2018; Fang et al., 2021b; Yao et al., 2022) and learnable threshold leak factors (Yin et al., 2021; 2023; Rathi & Roy, 2021). Additionally, bio-inspired approaches have incorporated multi-compartment structures to enhance neuronal dynamics for sequential modeling (Zhang et al., 2024; Chen et al., 2024; Hao et al., 2024) and time-series forecasting (Shibo et al., 2025). From the perspective of spike coding strategies, previous studies based on temporal coding (Yu et al., 2021; 2022) utilize augmented spikes to carry complementary information with spike coefficients in addition to spike latencies. Recent works based on rate coding expand binary spike trains into ternary spikes (Guo et al., 2024; Xing et al., 2024) or real-valued spikes (Guo et al., 2022c), while ensuring multiplication-free inference through the reparameterization technique. At the network structure level, recent works propose to enhance SNN architectures by redesigning the standard ResNet backbone (Fang et al., 2021a; Hu et al., 2024), and developing spike-driven transformers (Zhou et al., 2023; Yao et al., 2023a; 2024; Zhou et al., 2024; Yao et al., 2025). However, previous studies have shown limited improvements in spatiotemporal representational space and capacity of SNNs, as evidenced by our theoretical analysis in Section 3.5. These limitations inspire us to design a differentiable spatiotemporal transformation applied to the neuronal dynamics of SNNs to enhance their representations. Our proposed method leverages an input-distribution-aware and spatiotemporal-decoupled parametric strategy to overcome the limitations of binary representations in SNNs, while preserving the advantages of their event-driven and spike-based inference paradigm.

## 3 METHODOLOGY

### 3.1 PRELIMINARIES

**Spiking Neuron.** The Leaky Integrate-and-Fire (LIF) spiking neuron serves as the most popular fundamental building block of SNNs due to its low computational complexity (Maass, 1997). For implementation, the LIF neuron with soft reset is typically described in a discrete iterative format as follows:

$$\boldsymbol{u}_t^l = \lambda \boldsymbol{v}_{t-1}^l + \boldsymbol{W}^l \boldsymbol{s}_t^{l-1}, \tag{1}$$

$$\boldsymbol{s}_t^l = H\left(\boldsymbol{u}_t^l - \vartheta_{th}\right), \tag{2}$$

$$\boldsymbol{v}_t^l = \boldsymbol{u}_t^l - \vartheta_{th}\boldsymbol{s}_t^l, \tag{3}$$

where $\boldsymbol{u}_t^l$ and $\boldsymbol{s}_t^l$ denote the membrane potential and output spike of LIF neurons in layer $l$ at time step $t$, respectively. $\lambda$ is the decay factor of the membrane potential, and $\boldsymbol{W}^l$ represent the linear synaptic weights corresponding to layer $l$. $H(\cdot)$ is the Heaviside step function, defined as $H(x) = 1$ for $x \geq 0$ and $H(x) = 0$ for $x < 0$. When the membrane potential $\boldsymbol{u}_t^l$ exceeds its firing threshold $\vartheta_{th}$, the neuron will fire a spike to its post neurons and reset its membrane potential by subtracting $\vartheta_{th}\boldsymbol{s}_t^l$.

## 3.2 ANALYZING INFORMATION PROPAGATION FROM A TRANSFORMATION PERSPECTIVE

While the binary spike-based processing paradigm of SNNs provides computational efficiency, it induces performance degradation compared to full-precision counterparts. A critical bottleneck lies in the firing operation, which converts the full-precision membrane potential into binary spike trains that serve as information carriers propagating to the post-neurons. For notational simplicity, we omit layer superscripts in the following analysis. The information loss during the transformation of the full-precision membrane potential $\boldsymbol{u}$ into the binary spike form $\boldsymbol{s}$ through the firing operation is expressed as:

$$\mathcal{L} = \int_{t=1}^{T} d\left(\boldsymbol{s}\left(t\right), \boldsymbol{u}\left(t\right)\right) dt = \int_{t=1}^{T} d\left(g\left(\boldsymbol{u}\left(t\right)\right), \boldsymbol{u}\left(t\right)\right) dt, \tag{4}$$

where $g\left(\cdot\right)$ denotes the firing operation of spiking neurons, and $T$ signifies the total time steps. $d(\cdot, \cdot)$ represents a distance function used to measure the distance between two elements. Based on information theory, $d(\cdot, \cdot)$ can be interpreted as the mutual information function or the Kullback-Leibler (KL) divergence. In Euclidean space, $d(\cdot, \cdot)$ can alternatively be represented by the Frobenius norm.

Previous studies have focused on introducing learnable parameters (Fang et al., 2021b; Yao et al., 2022; Yin et al., 2021; 2023) and multi-compartment structures (Zhang et al., 2024; Chen et al., 2024; Hao et al., 2024; Shibo et al., 2025) to enhance the neuronal dynamics before the firing operation. These approaches can be regarded as adding functions with varying complexities before the firing operation. However, the reduction in information loss achieved by these methods is inherently limited, as they do not fundamentally expand the output space and capacity of SNNs (see our theoretical analysis in Section 3.5).

In this paper, we introduce a time-varying invertible transformation applied to neuronal dynamics in a conjugate manner, both before and after the firing operation. Mathematically, the corresponding output spikes of SNNs are augmented as:

$$\boldsymbol{s}\left(t\right) = f_t \circ g \circ f_t^{-1}\left(\boldsymbol{u}\left(t\right)\right), \tag{5}$$

where $f_t$ represents our introduced invertible transformation at time step $t$, and $\circ$ represents the composition operator for functions. The corresponding information loss caused by the firing operation is formulated as:

$$\int_{t=1}^{T} d\left(f_t \circ g \circ f_t^{-1}\left(\boldsymbol{u}\left(t\right)\right), \boldsymbol{u}\left(t\right)\right) dt. \tag{6}$$

Generally, the benefits of this design are twofold: (1) It provides additional flexibility to reduce firing errors by improving the neuronal dynamics before firing and expanding the spike representations after firing simultaneously. (2) This conjugate manner ensures variance consistency between the inputs and outputs, which is critical for stable information propagation in deep networks (He et al., 2015). Next, we will elaborate on how to parameterize the invertible transformation to expand the representational space of SNNs in a computationally efficient and input-distribution-aware manner.

## 3.3 INCORPORATING THE PARAMETRIC INVERTIBLE TRANSFORMATION INTO SNNS

In this section, we will elaborate on how to integrate our parametric invertible transformation (PIT) into SNNs. For the $l^{th}$ layer in the SNN, our introduced parametric invertible transformation $f_t^l$ can be expressed in a matrix form as:

$$f_t^l\left(\boldsymbol{u}_t^l\right) = \boldsymbol{A}_t^l\boldsymbol{u}_t^l, \tag{7}$$

where $\boldsymbol{A}_t^l \in \mathbb{R}^{N \times N}$ represents the transformation matrix in layer $l$ at time step $t$, and $\boldsymbol{u}_t^l \in \mathbb{R}^{B \times N}$ denotes the membrane potential of spiking neurons. $B$ and $N$ signify the batch size and hidden dimensions, respectively.

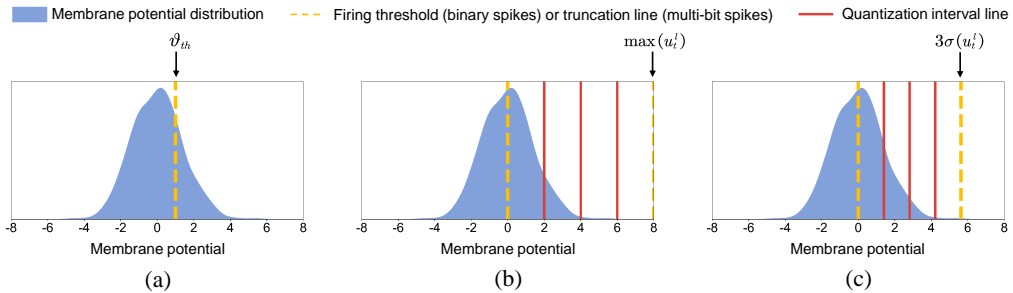

Figure 2: The comparison of different strategies for the membrane potential quantization. (a) The firing operation for the binary spiking neuron model. (b) The min-max quantization method for the multi-bit spiking neuron model. (c) Our quantization strategy is based on the 3-sigma rule of the normal distribution.

To reduce the information loss incurred by the inherent binary spikes of the LIF neuron model, we adopt the Integer Leaky Integrate-and-Fire (I-LIF) neuron (Luo et al., 2024), which allows emitting integer values during the training stage and maintains its spike-driven propagation during the inference stage through the reparameterization technique. Specifically, the firing dynamics of the I-LIF neuron can be rendered as:

$$s_t^l = \text{clip}\left(\lfloor u_t^l \rceil, 0, D\right), \tag{8}$$

where $\lfloor \cdot \rceil$ denotes the rounding operator. $\text{clip}(x, 0, D)$ confines the input $x$ within range $[0, D]$. And $D$ is a hyperparameter indicating the maximum integer value emitted by the I-LIF neuron.

As illustrated in Eq. (5), our proposed PIT is applied in a conjugate manner both before and after the firing operation. Collectively, incorporating our PIT into the I-LIF neuron yields the following neuronal dynamics in discrete form:

$$u_t^l = \lambda v_{t-1}^l + W^l s_t^{l-1}, \tag{9}$$

$$s_t^l = A_t^l \text{clip}\left(\lfloor \left(A_t^l\right)^{-1} u_t^l \rceil, 0, D\right), \tag{10}$$

$$v_t^l = u_t^l - s_t^l. \tag{11}$$

Eqs. (9), (10), and (11) correspond to the charging, firing, and resetting processes of the spiking neuron, respectively.

The next crucial question is how to parameterize our introduced PIT, i.e., the transformation matrix $A_t^l$ in Eq. (10), to expand the spatiotemporal representations of SNNs. For the parameterization strategy of the transformation matrix $A_t^l$, several aspects need to be considered: (1) The parameterization should be efficient in terms of both memory and computation. (2) It could dynamically adapt to the complex spatiotemporal distribution of the input data streams.

To address the first point, we design $A_t^l$ in a diagonal form, enabling efficient computation while introducing a negligible amount of additional parameters. Specifically, $A_t^l = \text{diag}\left(a_t^l\right)$, where $a_t^l \in \mathbb{R}^N$ and $N$ denotes the hidden dimension. In this way, the computation involved in the firing operation in Eq. (10) is simplified to element-wise production and division, which also facilitates preserving the event-driven and spike-based computational paradigm during the inference stage, as detailed in Appendix A. Further discussion about the structure of $A_t^l$ is also provided in Appendix F.

To tackle the second point, we propose an input-distribution-aware initialization strategy for $A_t^l$ based on the statistical distribution of the input data, instead of directly applying the min-max method (Xiao et al., 2023; Shao et al., 2024), which is heavily influenced by large outliers of inputs. As shown in Figure 2, the min-max strategy suffers from two major drawbacks: (1) It leads to an unstable quantization range, thereby slowing down the convergence of the model. (2) It wastes a significant portion of invaluable quantization levels, resulting in larger quantization errors, especially in ultra-low-bit scenarios, i.e., $D$ is small. Our approach mitigates these issues by designing a channel-wise initialization method based on the statistical distribution of the input tensor. Specifically, we initialize $a_t^l$ based on the 3-sigma rule of the normal distribution as:

$$a_t^l = \max(|\mu\left(u_t^l\right) - 3\sigma\left(u_t^l\right)|, |\mu\left(u_t^l\right) + 3\sigma\left(u_t^l\right)|)/\sqrt{D}, \tag{12}$$

where $\mu$ and $\sigma$ are calculated based on the mean and standard deviation of the membrane potential $u_t^l$ under the first batch of training data. After initialization, $a_t^l$ are updated according to BPTT (refer to Appendix B for the detailed learning rules).

### 3.4 RECTIFYING SURROGATE GRADIENTS DURING BACKPROPAGATION

The non-differentiable nature of the round function in Eq. (10) poses challenges for training SNNs using the BPTT algorithm. Previous studies have explored various surrogate gradient functions (Wu et al., 2018; Rathi & Roy, 2021; Li et al., 2021) to approximate derivatives during backpropagation for binary spikes. For integer spike forms, Luo et al. (Luo et al., 2024) employed a rectangular surrogate gradient function to train SNNs by retaining gradients for neurons activated within the $[0, D]$ range, masking all others. This approach resembles the straight-through estimator (STE) (Rosenblatt, 1957; Bengio et al., 2013), where the identity function is used as a proxy for the derivative of the rounding function. However, previous studies (Yin et al., 2019; Spallanzani et al., 2022; Wu et al., 2023b; Nagel et al., 2022) have demonstrated that directly using STE can lead to gradient mismatches, resulting in oscillation phenomena that hinder the network from converging to a local minimum with strong generalization ability.

Motivated by this, we rectify the gradient based on the distance between the input and the decision boundary of the rounding function. Specifically, the derivative of the rounding function with respect to the input is calculated in the following form:

$$\frac{\partial \lfloor x \rceil}{\partial x} = 1 + \lambda \left(0.5 - \text{sign}\left(dis(x)\right) dis(x)\right), \tag{13}$$

where $dis(x) = x - \lfloor x \rfloor - 0.5$ denotes the distance between the input $x$ and the decision boundary of the round function. $\lceil \cdot \rceil$ and $\text{sign}(\cdot)$ represent the ceiling function and sign function, respectively. Noting that the value of the distance variable falls within the range $[-0.5, 0.5]$. Specifically, the design of Eq. (13) can be deemed as adding an $\ell_2$ penalty on $\|0.5 - \text{sign}(dis(x)) \cdot dis(x)\|_2^2$. This regularization encourages the input to move away from the decision boundary, thereby mitigating oscillations of the output values between adjacent quantized states. As a result, the statistics of the rounding function's outputs are stabilized, reducing oscillation during training and facilitating convergence to a solution with stronger generalizability.

Collectively, by employing the BPTT algorithm along with our rectified surrogate gradient function, all learnable parameters of the model, including the introduced transformation matrix $\boldsymbol{A}_t^l$ in Eq. (10), can be effectively optimized (detailed learning rules for learnable parameters are provided in the Appendix B).

### 3.5 ANALYSIS OF SPATIOTEMPORAL REPRESENTATION SPACE OF SNNS

This section aims to analyze the spatiotemporal representation space of SNNs from a theoretical perspective. We first provide the mathematical definition of the representation space and capacity based on linear algebra, followed by a thorough comparison of the SNN embedded with the PIT and prior works.

**Definition 1.** *Given $\boldsymbol{s} = \{s_1, s_2, \cdots, s_N\}$, which refers to $N$ linearly independent elements, the corresponding representation space (denoted as $Span$) generated by it can be expressed as follows:*

$$Span\{\boldsymbol{s}\} = Span\{s_1, s_2, \cdots, s_N\} = \left\{\sum_{j=1}^{N} k_j s_j \mid k_j \in \mathbb{R}\right\}, \tag{14}$$

*where $k_j$ represents the linear combination coefficients (in SNNs, these are typically the weights in the linear layer or the convolution kernels in the convolutional layer).*

**Definition 2.** *Given $Span\{\boldsymbol{s}\} = \left\{\sum_{j=1}^{N} k_j s_j \mid k_j \in \mathbb{R}\right\}$, which denotes the representation space generated by $\boldsymbol{s}$ and $k$, the corresponding representation capacity (dubbed Cap) is evaluated as follows:*

$$Cap(Span\{\boldsymbol{s}\}) = \log \left|\left\{\sum_{j=1}^{N} k_j s_j \mid k_j \in \mathbb{R}, s_j \in \{0, 1, \ldots, D\}\right\}\right|, \tag{15}$$

*where $|\cdot|$ denotes the cardinality of the set, i.e., the total number of distinct elements generated by all possible combinations of $\boldsymbol{s}$ given $k$. In SNNs with multi-bit spiking neurons, $D$ refers to the quantization level of $\boldsymbol{s}$, and $N$ is the number of linearly independent elements in $\boldsymbol{s}$.*

**Proposition 1.** *For a SNN with multi-bit spiking neurons emitting the output spike trains $\{s_t\}_{t=1}^T = \{s_{ij} \mid s_{ij} \in \{0, \cdots, D\}, \forall j \in \{1, \cdots, N\}, i \in \{1, \cdots, T\}\}$, where $T$, $N$, and $D$ denote the total time step, hidden dimension, and quantization level, respectively, its corresponding spatiotemporal representation space and representation capacity have the following form:*

$$Span\{s_t\}_{t=1}^T = \left\{\sum_{i=1}^T \sum_{j=1}^N k_j s_{ij} \mid k_j \in \mathbb{R}, s_{ij} \in \{0, \cdots, D\}\right\}, \tag{16}$$

$$Cap(Span\{s_t\}_{t=1}^T) = \log(T \cdot (D+1)^N). \tag{17}$$

**Corollary 1.** *For a SNN with multi-bit spiking neurons, incorporating PIT generates the output spike trains $\{A_t s_t\}_{t=1}^T = \{a_{ij} s_{ij} \mid a_{ij} \in \mathbb{R}, s_{ij} \in \{0, \cdots, D\}\}$. The corresponding spatio-temporal representation space and representation capacity of the SNN embedded with PIT can be described as:*

$$Span\{A_t s_t\}_{t=1}^T = \left\{\sum_{i=1}^T \sum_{j=1}^N a_{ij} k_j s_{ij} \mid a_{ij} \in \mathbb{R}, k_j \in \mathbb{R}, s_{ij} \in \{0, \cdots, D\}\right\}, \tag{18}$$

$$Cap(Span\{A_t s_t\}_{t=1}^T) = \log(T \cdot (D+1)^N). \tag{19}$$

Proposition 1 elucidates the logarithmic relationship between the representation capacity and the quantization bit $D$ as well as the total time step $T$. Specifically, as the quantization bit $D$ increases, the representation capacity grows at a sublinear rate, while the capacity scales linearly with the hidden dimension $N$. Corollary 1 demonstrates that integrating additional parameters, i.e., $\{A_i\}_{i=1}^T$ in our proposed PIT, into the combination coefficients $\{k_j\}_{j=1}^N$ could increases the representation space of the SNN by offering more degrees of freedom for temporal and spatial variations in output signals. Detailed proofs of Proposition 1 and Corollary 1 are provided in Appendix C. Based on Proposition 1 and Corollary 1, Table 1 summarizes the comparative results of the spatiotemporal representation space and capacity between previous works and our proposed method.

Table 1: Comparison of the representation space and capacity of the currently developed SNNs

| Method | Representation Space | Representation Capacity |
|---|---|---|
| Vanilla Binary Spike | $\left\{\sum_{i=1}^T \sum_{j=1}^N k_j s_{ij} \mid k_j \in \mathbb{R}, s_{ij} \in \{0, 1\}\right\}$ | $\log(T \cdot 2^N)$ |
| Ternary Spike (Guo et al., 2024) | $\left\{\sum_{i=1}^T \sum_{j=1}^N k_j s_{ij} \mid k_j \in \mathbb{R}, s_{ij} \in \{-1, 0, 1\}\right\}$ | $\log(T \cdot 3^N)$ |
| Trainable Ternary Spike (Guo et al., 2024) | $\left\{\sum_{i=1}^T \sum_{j=1}^N a k_j s_{ij} \mid a \in \mathbb{R}, k_j \in \mathbb{R}, s_{ij} \in \{-1, 0, 1\}\right\}$ | $\log(T \cdot 3^N)$ |
| Real Spike (Guo et al., 2022c) | $\left\{\sum_{i=1}^T \sum_{j=1}^N a_j k_j s_{ij} \mid a_j \in \mathbb{R}, k_j \in \mathbb{R}, s_{ij} \in \{0, 1\}\right\}$ | $\log(T \cdot 2^N)$ |
| Ours | $\left\{\sum_{i=1}^T \sum_{j=1}^N a_{ij} k_j s_{ij} \mid a_{ij} \in \mathbb{R}, k_j \in \mathbb{R}, s_{ij} \in \{0, \cdots, D\}\right\}$ | $\log(T \cdot (D+1)^N)$ |

## 4 EXPERIMENTS

We conduct extensive experiments on CIFAR10 (Krizhevsky et al., 2009), CIFAR100 (Krizhevsky et al., 2009), ImageNet-1k (Deng et al., 2009), CIFAR10-DVS (Li et al., 2017), and DVS-Gesture (Amir et al., 2017) datasets to evaluate the performance of our proposed method across various architectures. The Pytorch (Paszke et al., 2019) and SpikingJelly (Fang et al., 2023) frameworks are utilized to implement SNN training in this paper. Detailed experimental setups are provided in Appendix D.2.

### 4.1 COMPARISON WITH THE STATE-OF-THE-ART

**Results on static image datasets.** We assess the effectiveness of our method on static image datasets, including CIFAR10 and CIFAR100, and ImageNet-1k. The results are summarized in Table 2. On the CIFAR10 dataset, integrating our proposed PIT into the baseline model achieves remarkable accuracy improvements compared to prior works, including maximizing the output information (IM-Loss (Guo et al., 2022a)) and rectifying the membrane potential distribution (RecDis-SNN (Guo et al., 2022b)). Furthermore, on the CIFAR100 dataset, with the introduction of PIT, the classification accuracy of the ResNet19 model surpasses prior methods designed to enhance spike representations, including

Table 2: Comparison with SOTA methods on CIFAR10, CIFAR100 and ImageNet-1k. Our results are reported as averages over three experimental runs with different random seeds. We reformulate the time steps for all direct training methods as $T \times D$, where $T$ represents the number of time steps and $D$ denotes the upper integer activation value of the firing function. In previous directing training work of SNNs, $D$ is set to 1 by default.

| Dataset | Method | Type | Architecture | Params | $T \times D$ | Accuracy |
|---|---|---|---|---|---|---|
| CIFAR10 | TL (Wu et al., 2023a) | Tandem learning | CIFARNet | 44.48M | $8 \times 1$ | 89.04% |
| | PTL (Wu et al., 2022) | Tandem learning | VGG11 | 9.23M | $16 \times 1$ | 91.24% |
| | PLIF (Fang et al., 2021b) | SNN training | PLIFNet | 37.31M | $8 \times 1$ | 93.50% |
| | DSR (Meng et al., 2022) | SNN training | ResNet18 | 11.18M | $20 \times 1$ | 95.40% |
| | RecDis-SNN (Guo et al., 2022b) | SNN training | ResNet19 | 12.70M | $2 \times 1$ | 93.64% |
| | IM-Loss (Guo et al., 2022a) | SNN training | ResNet19 | 12.70M | $4 \times 1$ | 95.40% |
| | Diet-SNN (Rathi & Roy, 2021) | SNN training | ResNet20 | 11.25M | $5 \times 1$ / $10 \times 1$ | 91.78% / 92.54% |
| | Dspike (Li et al., 2021) | SNN training | ResNet18 | 11.18M | $2 \times 1$ / $4 \times 1$ | 93.13% / 93.66% |
| | STBP-tdBN (Zheng et al., 2021) | SNN training | ResNet19 | 12.70M | $2 \times 1$ / $4 \times 1$ | 92.34% / 92.92% |
| | TET (Deng et al., 2022) | SNN training | ResNet19 | 12.70M | $2 \times 1$ / $4 \times 1$ | 94.16% / 94.44% |
| | Trainable Ternary Spike (Guo et al., 2024) | SNN training | ResNet19 | 12.70M | $1 \times 2$ / $2 \times 2$ | 95.58% / 95.80% |
| | Real Spike (Guo et al., 2022c) | SNN training | ResNet19 | 12.70M | $2 \times 1$ / $4 \times 1$ | 94.01% / 95.60% |
| | **PIT (Ours)** | SNN training | ResNet18 | 11.18M | $1 \times 2$ / $1 \times 4$ | **95.07%** $\pm$ 0.17 / **95.86%** $\pm$ 0.12 |
| | | | ResNet19 | 12.70M | $1 \times 2$ / $1 \times 4$ | **96.31%** $\pm$ 0.08 / **96.72%** $\pm$ 0.10 |
| CIFAR100 | QCFS (Bu et al., 2022) | ANN2SNN | ResNet20 | 11.25M | $64 \times 1$ | 70.49% |
| | LTL (Yang et al., 2022) | Tandem learning | ResNet20 | 11.25M | $31 \times 1$ | 76.08% |
| | Diet-SNN (Rathi & Roy, 2021) | SNN training | ResNet20 | 11.25M | $5 \times 1$ | 64.07% |
| | RecDis-SNN (Guo et al., 2022b) | SNN training | ResNet19 | 12.70M | $4 \times 1$ | 74.10% |
| | IM-Loss (Guo et al., 2022a) | SNN training | VGG16 | 14.72M | $5 \times 1$ | 70.18% |
| | Dspike (Li et al., 2021) | SNN training | ResNet18 | 11.18M | $2 \times 1$ / $4 \times 1$ | 71.68% / 73.35% |
| | TET (Deng et al., 2022) | SNN training | ResNet19 | 12.70M | $2 \times 1$ / $4 \times 1$ | 72.87% / 74.47% |
| | Trainable Ternary Spike (Guo et al., 2024) | SNN training | ResNet19 | 12.70M | $1 \times 2$ / $2 \times 2$ | 78.45% / 80.20% |
| | Real Spike (Guo et al., 2022c) | SNN training | ResNet20 / VGG16 | 12.70M / 14.72M | $5 \times 1$ / $5 \times 1$ | 66.60% / 70.62% |
| | **PIT (Ours)** | SNN training | ResNet18 | 11.18M | $1 \times 2$ / $1 \times 4$ | **76.88%** $\pm$ 0.12 / **78.83%** $\pm$ 0.10 |
| | | | ResNet19 | 12.70M | $1 \times 2$ / $1 \times 4$ | **80.12%** $\pm$ 0.10 / **81.59%** $\pm$ 0.09 |
| ImageNet-1k | STBP-tdBN (Zheng et al., 2021) | SNN training | ResNet34 | 21.79M | $6 \times 1$ | 63.72% |
| | TET (Deng et al., 2022) | SNN training | ResNet34 | 21.79M | $6 \times 1$ | 64.79% |
| | GLIF (Yao et al., 2022) | SNN training | ResNet34 | 21.79M | $4 \times 1$ | 67.52% |
| | DSR (Meng et al., 2022) | SNN training | ResNet18 | 11.69M | $50 \times 1$ | 67.74% |
| | TEBN (Duan et al., 2022) | SNN training | ResNet34 | 21.79M | $4 \times 1$ | 68.28% |
| | MS-ResNet (Hu et al., 2024) | SNN training | ResNet18 / ResNet34 | 11.69M / 21.79M | $6 \times 1$ / $6 \times 1$ | 63.10% / 69.42% |
| | Trainable Ternary Spike (Guo et al., 2024) | SNN training | ResNet18 / ResNet34 | 11.69M / 21.79M | $4 \times 2$ / $4 \times 2$ | 67.68% / 70.74% |
| | Real Spike (Guo et al., 2022c) | SNN training | ResNet18 / ResNet34 | 11.69M / 21.79M | $4 \times 1$ / $4 \times 1$ | 63.68% / 67.69% |
| | SEW ResNet (Fang et al., 2021a) | SNN training | ResNet18 | 11.69M | $4 \times 1$ | 63.18% |
| | | | ResNet34 | 21.79M | $4 \times 1$ | 67.04% |
| | | | ResNet50 | 25.56M | $4 \times 1$ | 67.78% |
| | | | ResNet101 | 44.55M | $4 \times 1$ | 68.76% |
| | | | ResNet152 | 60.19M | $4 \times 1$ | 69.26% |
| | Spike-Driven Transformer (Yao et al., 2023a) | SNN training | Spike-driven Transformer | 16.81M | $4 \times 1$ | 72.28% |
| | E-SpikeFormer (Yao et al., 2025) | SNN training | E-SpikeFormer-S | 5.11M | $1 \times 4$ | 75.30% |
| | | | E-SpikeFormer-M | 10.02M | $1 \times 4$ | 78.50% |
| | **PIT (Ours)** | SNN training | ResNet18 | 11.69M | $1 \times 4$ | **69.39%** $\pm$ 0.24 |
| | | | ResNet34 | 21.79M | $1 \times 4$ | **72.66%** $\pm$ 0.27 |
| | | | Spike-driven Transformer | 16.81M | $1 \times 4$ | **73.45%** $\pm$ 0.26 |
| | | | E-SpikeFormer-S | 5.11M | $1 \times 4$ | **76.00%** $\pm$ 0.22 |
| | | | E-SpikeFormer-M | 10.02M | $1 \times 4$ | **79.41%** $\pm$ 0.24 |

Real Spike (Guo et al., 2022c) and Trainable Ternary Spike (Guo et al., 2024), by 14.99% and 1.39%, respectively.

On the more challenging ImageNet-1k dataset, we incorporate our proposed PIT into the standard SEW ResNet (Fang et al., 2021a) architecture and compare it with state-of-the-art methods, including BN-based approaches (tdBN (Zheng et al., 2021), TEBN (Duan et al., 2022)) and spike representation enhancement methods (Real Spike (Guo et al., 2022c) and Trainable Ternary Spike (Guo et al., 2024)). As reported in Table 2, by applying PIT to the SEW ResNet18 and SEW ResNet34 models, we achieve accuracy improvements of 6.21% and 5.62% over the baselines, respectively. Notably, the SEW ResNet34 integrated with PIT surpasses the classification accuracy of the 152-layer SEW ResNet model, demonstrating the effectiveness and superiority of our method.

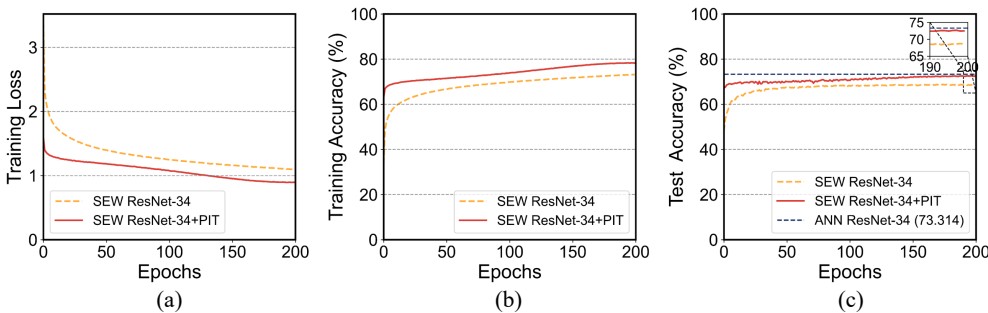

Figure 3: Comparison of (a) training loss, (b) training accuracy, and (c) test accuracy on ImageNet-1k.

**Results on neuromorphic datasets.** Table 11 and Table 12 in Appendix E present the experimental results on the CIFAR10-DVS and DVS-Gesture datasets, respectively. On the CIFAR10-DVS dataset, when utilizing VGG11 under equivalent inference latency ($T \times D = 10$), our method surpasses previous approaches aimed at improving spike representation, including Real Spike (Guo et al., 2022c) and Trainable Ternary Spike (Guo et al., 2024), in terms of accuracy by 5.92% and 3.90%, respectively. Moreover, as the quantization level $D$ increases, the model's classification performance can be further improved. Similar conclusions can be drawn from the comparison in Table 12 on the DVS-Gesture dataset.

## 4.2 ANALYSIS OF OPTIMIZATION DYNAMICS

**Visualization of learning curves.** To investigate the optimization procedure during the training stage, we plot the learning curves of our methods and their vanilla counterparts on ImageNet-1k in Figure 3. It is evident that the introduction of PIT facilitates rapid model convergence and ultimately achieves higher performance. Specifically, the SEW ResNet34 integrated with PIT (red line) converges rapidly and eventually achieves a superior accuracy level (72.66%), closely matching its ANN counterpart (ResNet34 (He et al., 2016)), which achieves 73.31%.

**Visualization of membrane potential distribution.** We visualize the membrane potential distribution after the last convolutional layer of VGG11 for vanilla LIF, I-LIF (Luo et al., 2024), and our PIT on CIFAR10-DVS, as shown in Figure 4. The surface plot is smoothed using Gaussian filtering for better visualization. We observe that vanilla LIF exhibits minimal discriminative ability for inputs at different timesteps, showing uniform distributions along the temporal dimension. In contrast, both I-LIF and our PIT demonstrate better temporal resolution. Notably, our PIT dynamically adjusts the potential distribution for inputs at different timesteps, particularly at $T = 3$ and $T = 4$, highlighting the ability of our method to better capture the temporal structure of the data stream.

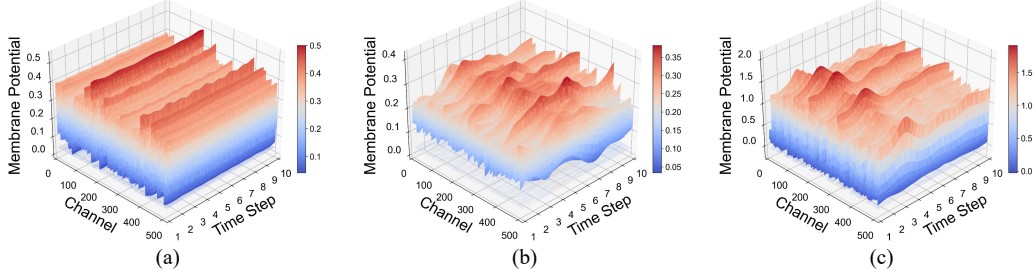

Figure 4: Comparison of the membrane potential distribution after the last convolutional layer of VGG11 for (a) vanilla LIF, (b) I-LIF, and (c) PIT on CIFAR10-DVS.

Table 3: Energy cost comparison on CIFAR10 and CIFAR100.

| Dataset | Architecture | Method | $T \times D$ | S-ACE (G) | NS-ACE (G) | Params (M) | Accuracy |
|---------|--------------|--------|--------------|-----------|------------|-----------|----------|
| CIFAR10 | ResNet18 | LIF | $4 \times 1$ | 35.84 | 6.45 | 11.18 | 94.22% |
| | ResNet18 | QAT | N / A | 35.84 | - | 11.18 | 95.14% |
| | ResNet18 | PIT | $1 \times 4$ | 35.84 | 4.31 | 11.18 | 95.86% |
| CIFAR100 | ResNet18 | LIF | $4 \times 1$ | 35.84 | 7.88 | 11.18 | 73.25% |
| | ResNet18 | QAT | N / A | 35.84 | - | 11.18 | 77.91% |
| | ResNet18 | PIT | $1 \times 4$ | 35.84 | 5.73 | 11.18 | 78.83% |

## 5 ABLATION STUDIES

In this section, we conduct comprehensive ablation studies to investigate the individual contributions of the core components within our proposed framework and evaluate its computational overhead.

**Sensitivity to hyperparameter $\lambda$ in the rectified surrogate gradient function.** Table 4 presents the sensitivity analysis of $\lambda$ in our rectified surrogate gradient function (Eq. (13)). We vary $\lambda$ from 0.1 to 0.001 and observe consistent performance across this range, with accuracy variations less than 1% on CIFAR10 and CIFAR100. The stable performance within the broad range of [0.001, 0.1] indicates that our gradient correction mechanism is robust to hyperparameter selection, eliminating the need for extensive tuning in practical applications.

Table 4: Impact of hyperparameter $\lambda$ in the rectified surrogate gradient function.

| Dataset | Architecture | Params | $T \times D$ | $\lambda$ | Accuracy |
|---|---|---|---|---|---|
| CIFAR10 | ResNet18 | 11.18M | $1 \times 4$ | 0.1 | 95.73% |
| | ResNet18 | 11.18M | $1 \times 4$ | 0.01 | 95.86% |
| | ResNet18 | 11.18M | $1 \times 4$ | 0.001 | 95.70% |
| CIFAR100 | ResNet18 | 11.18M | $1 \times 4$ | 0.1 | 78.10% |
| | ResNet18 | 11.18M | $1 \times 4$ | 0.01 | 78.83% |
| | ResNet18 | 11.18M | $1 \times 4$ | 0.001 | 79.01% |

**Memory overhead analysis.** Table 5 compares the training memory footprint of our method against baseline SNNs. On CIFAR10, our approach incurs modest memory overhead (0.22 GB and 0.25 GB for ResNet18 and ResNet19, respectively) while achieving substantial accuracy gains of 1.64% and 1.23%. On ImageNet-1k, despite requiring additional memory, our method delivers significant performance improvements of 6.21% and 5.62% compared to SEW ResNet18 and SEW ResNet34 (Fang et al., 2021a), respectively. This favorable trade-off between performance and memory demonstrates the practical viability of our approach.

Table 5: Memory overhead comparison during training.

| Dataset | Architecture | Model | Params | $T \times D$ | Memory | Accuracy |
|---|---|---|---|---|---|---|
| CIFAR10 | ResNet18 | LIF | 11.18M | $4 \times 1$ | 2.21 GB | 94.22% |
| | ResNet19 | LIF | 12.70M | $4 \times 1$ | 4.45 GB | 95.49% |
| | ResNet18 | PIT | 11.18M | $1 \times 4$ | 2.43 GB | 95.86% |
| | ResNet19 | PIT | 12.70M | $1 \times 4$ | 4.70 GB | 96.72% |
| ImageNet-1k | SEW ResNet18 | IF | 11.69M | $4 \times 1$ | 13.65 GB | 63.18% |
| | SEW ResNet34 | IF | 21.79M | $4 \times 1$ | 18.86 GB | 67.04% |
| | SEW ResNet18 | PIT | 11.69M | $1 \times 4$ | 17.87 GB | 69.39% |
| | SEW ResNet34 | PIT | 21.79M | $1 \times 4$ | 24.66 GB | 72.66% |

**Energy efficiency analysis.** Following Shen et al. (2024), we evaluate energy efficiency using Synaptic Arithmetic Computation Effort (S-ACE) and Neuromorphic Synaptic Arithmetic Computation Effort (NS-ACE) metrics. Table 3 reports comparisons with vanilla SNNs and Quantization Aware Training (QAT) baseline that uses the same firing count and per-channel scale learning as in our PIT. The S-ACE and NS-ACE are computed under 16-bit weight precision. At the same S-ACE budgets, our method consistently outperforms both LIF and QAT baselines, with accuracy improvements of 1.64% and 0.72% on CIFAR10, and 5.58% and 0.92% on CIFAR100. Notably, our approach achieves lower NS-ACE than SNNs with vanilla LIF, which can be attributed to the adaptive modulation mechanism of PIT that reduces firing activity. These results demonstrate that our method simultaneously improves both accuracy and energy efficiency.

## 6 DISCUSSION AND COMPARISON TO EXISTING WORKS

This paper analyzes information propagation in SNNs from a continuous-valued transformation perspective, offering a fundamental understanding beyond conventional ANN-to-SNN conversion approaches. We provide three key insights for the community: (1) Mitigating information loss requires adaptively adjusting the firing mechanism based on the time-evolving membrane potential distribution and is insufficient when merely applying batch normalization to input currents (Kim & Panda, 2021; Duan et al., 2022). (2) We propose an auxiliary gradient correction term that adaptively addresses gradient mismatch and oscillation, surpassing pre-defined surrogate functions in facilitating convergence and generalization. (3) The inherent temporal dynamics of SNNs necessitate time-varying and heterogeneous methods that consider spatiotemporal dimensions, rather than the homogeneous strategies commonly adopted in the quantization literature (Esser et al., 2020).

## 7 CONCLUSION

In this work, we introduced a parametric invertible transformation into neuronal dynamics to enhance the spatiotemporal representations of SNNs. By further incorporating an input-distribution-aware parametric strategy and a rectified surrogate gradient function into SNNs, we demonstrated state-of-the-art performance across a broad range of tasks. Our theoretical analysis further provides insights into the enhanced spatiotemporal representations of our approach, paving the way for low-latency and high-accuracy neuromorphic computing systems.

## ACKNOWLEDGEMENT

This work was partially supported by the National Natural Science Foundation of China (Grant No. 62306259), the Research Grants Council of the Hong Kong SAR (Grant No. C5052-23G, PolyU25216423, and PolyU15217424), and the Hong Kong Polytechnic University (P0058445).

## ETHICS STATEMENT

We acknowledge and adhere to the ICLR Code of Ethics in all aspects of this research. This paper focuses on analyzing and expanding spatiotemporal representations of SNNs. In general, there are no direct negative social impacts associated with this work. The SNNs studied in this paper are dedicated to operating in an event-driven and spike-based computational paradigm, offering significant potential for energy efficiency and contributing to the reduction of carbon dioxide emissions. The method proposed in this work is applied to neuronal dynamics in a conjugate manner, enhancing the SNNs' spatiotemporal representations while preserving the advantages of the spike-driven inference computational paradigm. This work provides a theoretical foundation for analyzing and improving the spatiotemporal representation space and capacity of SNNs.

## REPRODUCIBILITY STATEMENT

To ensure the reproducibility of our work, we provide comprehensive documentation of all experimental and theoretical components. The complete descriptions of datasets used in our experiments, including detailed data processing steps and training configurations, are provided in Appendix D. The learning rules for our PIT method during backpropagation are thoroughly explained in Appendix B. Complete proofs for all theoretical claims and theorems presented in the main text can be found in Appendix C.

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

# Appendix

## A   IMPLEMENTATION OF SPIKE-DRIVEN INFERENCE

In this section, we illustrate the equivalence between real-valued training and binary spike-driven inference using the reparameterization technique. This enables the SNN integrated with our PIT to retain the advantages of spike-driven inference.

**Reparameterization technique.** Due to the introduction of our designed PIT, the output of spiking neurons (as illustrated in Eq. (10)) takes a real-valued form. This results in the multiplication of weights and activations in the SNN being converted into MAC operations instead of AC operations, thereby losing the computational efficiency advantage of SNNs. To address this issue, we follow a training-inference decoupled technique (Guo et al., 2022c; 2024), which converts the real-valued outputs into binary spikes during the inference stage while maintaining computational efficiency through weight folding.

The key insight is that the PIT matrix $\boldsymbol{A}_t^{l-1}$ can be folded into the weight matrix of the next layer through a reparameterization technique similar to that employed in Batch Normalization (Ioffe & Szegedy, 2015). This folding operation enables the preservation of spike-driven computation while accommodating the real-valued intermediate representations during training.

Here, we take a fully connected layer as an example to illustrate the reparameterization technique. For a fully connected layer, the inputs from the preceding layer are multiplied by a weight matrix to produce the output features, which can be expressed as follows:

$$\boldsymbol{O}_t^l = \boldsymbol{W}^l \boldsymbol{s}_t^{l-1}, \tag{20}$$

where $\boldsymbol{O}_t^l$ denotes the outputs of layer $l$ at time step $t$, and $\boldsymbol{W}^l$ represents the weight matrix associated with layer $l$. For standard SNNs, the inputs $\boldsymbol{s}_t^{l-1}$ consists of binary values. While in our model, the inputs are augmented by our introduced PIT, thus possessing the real-valued form. Recall the firing operation in Eq. (10), we have:

$$\boldsymbol{s}_t^{l-1} = \boldsymbol{A}_t^{l-1}\text{clip}\left(\lfloor\left(\boldsymbol{A}_t^{l-1}\right)^{-1}\boldsymbol{u}_t^{l-1}\rceil, 0, D\right) = \boldsymbol{A}_t^{l-1}\sum_{d=1}^{D}\hat{\boldsymbol{s}}_d^{l-1}, \tag{21}$$

where $\hat{\boldsymbol{s}}_d^{l-1}$ denotes the output spikes generated by the vanilla Integrate-and-Fire neuron model with Soft Reset (IF-SR) at time step $d$. Specifically, this involves feeding the IF-SR with the input $\left(\boldsymbol{A}_t^{l-1}\right)^{-1}\boldsymbol{u}_t^{l-1}$ at the first time step, and subsequently subtracting 1 at each following time step until $D$. This temporal dimension expanding strategy has also been validated by prior works (Luo et al., 2024; Yao et al., 2025).

In this manner, the calculation process during inference of Eq. (20) can be illustrated as follows:

$$\boldsymbol{O}_t^l = \boldsymbol{W}^l \boldsymbol{s}_t^{l-1} = \boldsymbol{W}^l\left(\boldsymbol{A}_t^{l-1}\sum_{d=1}^{D}\hat{\boldsymbol{s}}_d^{l-1}\right) = \left(\boldsymbol{W}^l\boldsymbol{A}_t^{l-1}\right)\sum_{d=1}^{D}\hat{\boldsymbol{s}}_d^{l-1}. \tag{22}$$

By folding our PIT matrix, i.e., $\boldsymbol{A}_t^{l-1}$, into the original weight matrix ($\boldsymbol{W}^l$) of the SNN, the whole model could maintain the spike-driven computational paradigm during the inference stage.

**Layer-specific implementation details.** The folding mechanism varies depending on the layer architecture, and we provide specific implementations for the most commonly used layer types in SNNs.

1. *Fully connected layers:* For a fully connected layer, the weight matrix of the $l$-th layer is $\boldsymbol{W}^l \in \mathbb{R}^{M\times N}$, where $N$ and $M$ denote the input and output dimensions, respectively. Since $\boldsymbol{A}_t^{l-1} \in \mathbb{R}^{N\times N}$, the product $\boldsymbol{W}^l\boldsymbol{A}_t^{l-1} \in \mathbb{R}^{M\times N}$ can be treated as a new weight matrix $\tilde{\boldsymbol{W}}^l \in \mathbb{R}^{M\times N}$, ensuring the spike-based inference paradigm. Importantly, $\boldsymbol{A}_t^{l-1}$ can be folded into the original weight matrix regardless of whether it is diagonalized, providing flexibility in the PIT design.

2. *Convolutional layers:* For convolutional layers, assume that the convolution kernel of the $l$-th layer is $\boldsymbol{W}^l \in \mathbb{R}^{C_{in}\times K_w\times K_h\times C_{out}}$, where $C_{in}$ and $C_{out}$ denote the input and output channels,

respectively, and $K_w$ and $K_h$ represent the kernel width and height. In practice, $\boldsymbol{A}_t^{l-1}$ is typically set to a diagonal form for computational efficiency, i.e., $\boldsymbol{A}_t^{l-1} = \text{diag}(\boldsymbol{a}_t^{l-1})$, where $\boldsymbol{a}_t^{l-1} \in \mathbb{R}^{C_{in}}$. To fold the PIT matrix into the convolution kernel, the diagonal elements $\boldsymbol{a}_t^{l-1}$ are absorbed into the channel dimension of $\boldsymbol{W}^l$ via the Hadamard product:

$$\tilde{\boldsymbol{W}}_{i,:,:,j}^l = a_{t,i}^{l-1} \cdot \boldsymbol{W}_{i,:,:,j}^l, \quad \forall i \in \{1, 2, \ldots, C_{in}\}, j \in \{1, 2, \ldots, C_{out}\}, \tag{23}$$

where $\tilde{\boldsymbol{W}}^l$ represents the folded weight matrix and $a_{t,i}^{l-1}$ denotes the $i$-th element of $\boldsymbol{a}_t^{l-1}$.

This reparameterization ensures that during inference, the network operates with binary spikes while preserving the representational space enhanced by the PIT during training. The folded weights $\tilde{\boldsymbol{W}}^l$ encapsulate both the original synaptic weights and the learned PIT parameters, enabling efficient spike-driven computation without sacrificing the benefits of real-valued training.

# B DETAILED DERIVATIONS OF LEARNING RULES DURING BACKPROPAGATION

By utilizing the BPTT algorithm (Wu et al., 2018; Neftci et al., 2019; Zenke & Vogels, 2021; Lee et al., 2020), the gradients of weight parameters in the SNN can be calculated as follows:

$$\frac{\partial \mathcal{L}}{\partial \boldsymbol{W}^l} = \sum_{t=1}^T \frac{\partial \mathcal{L}}{\partial \boldsymbol{s}_t^l} \frac{\partial \boldsymbol{s}_t^l}{\partial \boldsymbol{u}_t^l} \frac{\partial \boldsymbol{u}_t^l}{\partial \boldsymbol{W}} = \sum_{t=1}^T \frac{\partial \mathcal{L}}{\partial \boldsymbol{s}_t^l} \frac{\partial \boldsymbol{s}_t^l}{\partial \boldsymbol{u}_t^l} \boldsymbol{s}_t^{l-1}. \tag{24}$$

Recall the neuronal dynamics of the spiking neuron model integrated with PIT, which can be described in discrete form as follows:

$$\boldsymbol{u}_t^l = \lambda \boldsymbol{v}_{t-1}^l + \boldsymbol{W}^l \boldsymbol{s}_t^{l-1}, \tag{25}$$

$$\boldsymbol{s}_t^l = \boldsymbol{A}_t^l \text{clip}\left(\lfloor (\boldsymbol{A}_t^l)^{-1} \boldsymbol{u}_t^l \rceil, 0, D\right), \tag{26}$$

$$\boldsymbol{v}_t^l = \boldsymbol{u}_t^l - \boldsymbol{s}_t^l. \tag{27}$$

According to the firing operation illustrated in Eq. (26), we have:

$$
\begin{aligned}
\frac{\partial \boldsymbol{s}_t^l}{\partial \left((\boldsymbol{A}_t^l)^{-1} \boldsymbol{u}_t^l\right)} &= \frac{\partial \left(\boldsymbol{A}_t^l \text{clip}\left(\lfloor (\boldsymbol{A}_t^l)^{-1} \boldsymbol{u}_t^l \rceil, 0, D\right)\right)}{\partial \left((\boldsymbol{A}_t^l)^{-1} \boldsymbol{u}_t^l\right)} \\
&= \boldsymbol{A}_t^l \frac{\partial \lfloor (\boldsymbol{A}_t^l)^{-1} \boldsymbol{u}_t^l \rceil}{\partial \left((\boldsymbol{A}_t^l)^{-1} \boldsymbol{u}_t^l\right)} \text{ sign}\left(0 \le (\boldsymbol{A}_t^l)^{-1} \boldsymbol{u}_t^l \le D\right),
\end{aligned}
\tag{28}
$$

where $\text{sign}(\cdot)$ denotes the sign function. To overcome the non-differentiable nature of the round function in Eq. (28), we calculate its derivative based on the distance between the input and the decision boundary of the round function (as illustrated in Eq. (13)):

$$\frac{\partial \lfloor (\boldsymbol{A}_t^l)^{-1} \boldsymbol{u}_t^l \rceil}{\partial \left((\boldsymbol{A}_t^l)^{-1} \boldsymbol{u}_t^l\right)} = 1 + \lambda \left(0.5 - \text{sign}\left(dis\left((\boldsymbol{A}_t^l)^{-1} \boldsymbol{u}_t^l\right)\right) dis\left((\boldsymbol{A}_t^l)^{-1} \boldsymbol{u}_t^l\right)\right), \tag{29}$$

where $dis\left((\boldsymbol{A}_t^l)^{-1} \boldsymbol{u}_t^l\right) = (\boldsymbol{A}_t^l)^{-1} \boldsymbol{u}_t^l - \lfloor (\boldsymbol{A}_t^l)^{-1} \boldsymbol{u}_t^l \rfloor - 0.5$ that refers to the distance between the input and the decision boundary of the round function. $\lfloor \cdot \rfloor$ represents the ceiling function. $\lambda$ denotes the hyperparameter that controls the strength of the auxiliary rectifying term, which is set to $0.01$ in our experiments. Noting that the value of the distance variable $dis$ falls within the range $[-0.5, 0.5]$.

Thus, combining Eqs. (28) and (29), we could deduce:

$$
\begin{aligned}
\frac{\partial \boldsymbol{s}_t^l}{\partial \boldsymbol{u}_t^l} &= \frac{\partial \boldsymbol{s}_t^l}{\partial \left((\boldsymbol{A}_t^l)^{-1} \boldsymbol{u}_t^l\right)} \frac{\partial \left((\boldsymbol{A}_t^l)^{-1} \boldsymbol{u}_t^l\right)}{\partial \boldsymbol{u}_t^l} \\
&= \left(1 + \lambda \left(0.5 - \text{sign}\left(dis\left((\boldsymbol{A}_t^l)^{-1} \boldsymbol{u}_t^l\right)\right) dis\left((\boldsymbol{A}_t^l)^{-1} \boldsymbol{u}_t^l\right)\right)\right) \text{sign}\left(0 \le (\boldsymbol{A}_t^l)^{-1} \boldsymbol{u}_t^l \le D\right).
\end{aligned}
\tag{30}
$$

Regarding the gradient of learnable parameters (i.e., $\boldsymbol{A}_t^l$) involved in our introduced PIT, we could calculate it according to the chain rule as follows:

$$\frac{\partial \mathcal{L}}{\partial \boldsymbol{A}_t^l} = \frac{\partial \mathcal{L}}{\partial \boldsymbol{s}_t^l} \frac{\partial \boldsymbol{s}_t^l}{\partial \boldsymbol{A}_t^l}, \tag{31}$$

$$
\begin{aligned}
\frac{\partial \boldsymbol{s}_t^l}{\partial \boldsymbol{A}_t^l} &= \text{clip}\left(\lfloor (\boldsymbol{A}_t^l)^{-1}\, \boldsymbol{u}_t^l \rceil, 0, D\right) + \boldsymbol{A}_t^l \frac{\partial \left(\text{clip}\left(\lfloor (\boldsymbol{A}_t^l)^{-1}\, \boldsymbol{u}_t^l \rceil, 0, D\right)\right)}{\boldsymbol{A}_t^l} \\
&= \text{clip}\left(\lfloor (\boldsymbol{A}_t^l)^{-1}\, \boldsymbol{u}_t^l \rceil, 0, D\right) - \left((\boldsymbol{A}_t^l)^{-1}\, \boldsymbol{u}_t^l\right) \text{sign}\left(0 \le (\boldsymbol{A}_t^l)^{-1}\, \boldsymbol{u}_t^l \le D\right).
\end{aligned}
\tag{32}
$$

Collectively, all learnable parameters in our model could be trained in an end-to-end manner by utilizing the BPTT algorithm along with our designed surrogate gradient function.

## C  PROOF OF THEOREMS

**Proposition 1.** *For a SNN with multi-bit spiking neurons emitting the output spike trains $\{\boldsymbol{s}_t\}_{t=1}^T = \{s_{ij} \mid s_{ij} \in \{0, \cdots, D\}, \forall j \in \{1, \cdots, N\}, i \in \{1, \cdots, T\}\}$, where $T$, $N$, and $D$ denote the total time step, hidden dimension, and quantization level, respectively, its corresponding spatiotemporal representation space and representation capacity have the following form:*

$$Span\{\boldsymbol{s}_t\}_{t=1}^T = \left\{ \sum_{i=1}^T \sum_{j=1}^N k_j s_{ij} \mid k_j \in \mathbb{R}, s_{ij} \in \{0, \cdots, D\} \right\}, \tag{33}$$

$$Cap(Span\{\boldsymbol{s}_t\}_{t=1}^T) = \log(T \cdot (D+1)^N). \tag{34}$$

*Proof.* For a SNN with multi-bit spiking neurons, at each time step $t$, its output spike $\boldsymbol{s}_t$ consists of $N$ elements, each denoted as $s_{tj}$, where $j \in \{1, \cdots, N\}$. According to Definition 1, the representation space generated by $N$ linearly independent elements of $\boldsymbol{s}_t$ has the following form:

$$Span\{\boldsymbol{s}_t\} = \left\{ \sum_{j=1}^N k_j s_{tj} \mid k_j \in \mathbb{R}, s_{tj} \in \{0, \cdots, D\} \right\}. \tag{35}$$

Over $T$ time steps, the spike trains $\{\boldsymbol{s}_t\}_{t=1}^T$ are aggregated. The spatiotemporal representation space is therefore constructed by summing across all $T$ time steps:

$$Span\{\boldsymbol{s}_t\}_{t=1}^T = \left\{ \sum_{i=1}^T \sum_{j=1}^N k_j s_{ij} \mid k_j \in \mathbb{R}, s_{ij} \in \{0, \cdots, D\} \right\}. \tag{36}$$

Here, the inner summation $\sum_{j=1}^N k_j s_{ij}$ represents the contribution of the $N$ dimensions at time step $i$. The outer summation $\sum_{i=1}^T$ aggregates the contributions across $T$ time steps. This establishes the form of the spatiotemporal representation space in Eq. (33).

Next, we measure its representation capacity based on Definition 2. Specifically, we first calculate the cardinality (dubbed $Card$) of its representation space, i.e., $Span\{\boldsymbol{s}_t\}_{t=1}^T$, which yields:

$$Card(Span\{\boldsymbol{s}_t\}_{t=1}^T) = \left| \left\{ \sum_{i=1}^T \sum_{j=1}^N k_j s_{ij} \mid k_j \in \mathbb{R}, s_{ij} \in \{0, \cdots, D\} \right\} \right|. \tag{37}$$

Given coefficients $\{k_j\}_{j=1}^N$ in linear layers or convolutional kernels of the SNN, the cardinality of the representation space is determined by the distinct values that $\{s_{ij}\}$ can take, given their quantization level $D$. Each $s_{ij}$ is quantized to $D+1$ discrete levels: $\{0, 1, \cdots, D\}$. Across $N$ dimensions, the total number of possible combinations of $\{s_{ij}\}_{j=1}^N$ at a single time step $i$ is $(D+1)^N$. Over $T$ time steps, each time step introduces a new combination of $\{s_{ij}\}_{j=1}^N$. Thus, the total number of distinct configurations across $T$ time steps is: $T \cdot (D+1)^N$.

Finally, taking the logarithm of the cardinality, the corresponding representation capacity is:

$$Cap(Span\{\boldsymbol{s}_t\}_{t=1}^T) = \log\left(T \cdot (D+1)^N\right). \tag{38}$$

This establishes the expression for the representation capacity illustrated in Eq. (34).

**Corollary 1.** *For a SNN with multi-bit spiking neurons, incorporating PIT generates the output spike trains $\{\boldsymbol{A}_t\boldsymbol{s}_t\}_{t=1}^T = \{a_{ij}s_{ij} \mid a_{ij} \in \mathbb{R}, s_{ij} \in \{0, \cdots, D\}\}$. The corresponding spatio-temporal representation space and representation capacity of the SNN embedded with PIT can be described as:*

$$Span\left\{\boldsymbol{A}_t\boldsymbol{s}_t\right\}_{t=1}^T = \left\{\sum_{i=1}^T\sum_{j=1}^N a_{ij}k_j s_{ij} \mid a_{ij} \in \mathbb{R}, k_j \in \mathbb{R}, s_{ij} \in \{0, \cdots, D\}\right\}, \tag{39}$$

$$Cap(Span\left\{\boldsymbol{A}_t\boldsymbol{s}_t\right\}_{t=1}^T) = \log(T \cdot (D+1)^N). \tag{40}$$

*Proof.* For a SNN with multi-bit spiking neurons, our introduced PIT extends the spike train $\{\boldsymbol{s}_t\}_{t=1}^T$ by introducing parameters $a_{ij} \in \mathbb{R}$, which are applied to each element $s_{ij}$. The resulting transformed spike trains are:

$$\{\boldsymbol{A}_t\boldsymbol{s}_t\}_{t=1}^T = \{a_{ij}s_{ij} \mid a_{ij} \in \mathbb{R}, s_{ij} \in \{0, \cdots, D\}, i \in \{1, \cdots, T\}, j \in \{1, \cdots, N\}\}. \tag{41}$$

PIT scales each spike value $s_{ij}$ by a real-valued parameter $a_{ij}$, effectively expanding the representation space by introducing additional degrees of freedom across spatial and temporal dimensions.

Based on Definition 1, aggregating over $T$ time steps, its corresponding spatio-temporal representation space of the model becomes:

$$Span\left\{\boldsymbol{A}_t\boldsymbol{s}_t\right\}_{t=1}^T = \left\{\sum_{i=1}^T\sum_{j=1}^N a_{ij}k_j s_{ij} \mid a_{ij} \in \mathbb{R}, k_j \in \mathbb{R}, s_{ij} \in \{0, \cdots, D\}\right\}. \tag{42}$$

This establishes the form of the spatio-temporal representation space in Eq. (39).

According to Definition 2, its representation capacity is determined by the logarithmic scale of the number of distinct elements in the set $Span\left\{\boldsymbol{A}_t\boldsymbol{s}_t\right\}_{t=1}^T$:

$$Cap(Span\left\{\boldsymbol{A}_t\boldsymbol{s}_t\right\}_{t=1}^T) = \log\left|\left\{\sum_{i=1}^T\sum_{j=1}^N a_{ij}k_j s_{ij} \mid a_{ij} \in \mathbb{R}, k_j \in \mathbb{R}, s_{ij} \in \{0, \cdots, D\}\right\}\right|. \tag{43}$$

Given introduced parameters $a_{ij}$ in PIT and $k_j$ in linear layers or convolutional kernels of the SNN, the cardinality of the representation space is determined by the distinct values that $s_{ij}$ can take, given their quantization level $D$. Each $s_{ij}$ is quantized to $D+1$ discrete levels: $\{0, 1, \cdots, D\}$. Across $N$ dimensions, the total number of possible combinations of $\{s_{ij}\}_{j=1}^N$ at a single time step $i$ is $(D+1)^N$. Over $T$ time steps, each time step introduces a new combination of $\{s_{ij}\}_{j=1}^N$. Thus, the total number of distinct configurations across $T$ time steps is $T \cdot (D+1)^N$.

Finally, taking the logarithm of the cardinality of the representation space, its representation capacity yields:

$$Cap(Span\left\{\boldsymbol{A}_t\boldsymbol{s}_t\right\}_{t=1}^T) = \log\left(T \cdot (D+1)^N\right). \tag{44}$$

This establishes the expression for the representation capacity in Eq. (40).

## D EXPERIMENTAL SETTINGS

### D.1 DATASETS

**CIFAR-10 and CIFAR-100.** The CIFAR-10 and CIFAR-100 datasets (Krizhevsky et al., 2009) consist of 32x32 color images categorized into multiple classes. Specifically, CIFAR-10 comprises 60,000 images distributed across 10 classes, with 50,000 images allocated for training and 10,000 for testing, while CIFAR-100 contains images spanning 100 distinct classes. Both datasets have

been preprocessed to achieve zero mean and unit variance. Image data augmentation techniques, including AutoAugment (Cubuk et al., 2019) and Cutout (DeVries & Taylor, 2017), are employed, following the methodologies outlined in prior studies (Guo et al., 2022a; Bu et al., 2022; Wang et al., 2023). The pixel values are directly input into the model's input layer at each timestep using a direct encoding method (Rathi & Roy, 2021). Following previous works (Huang et al., 2024; Zheng et al., 2021; Deng et al., 2022), the Spiking ResNet18 and ResNet19 are utilized as backbone models for CIFAR-10 and CIFAR-100.

**ImageNet-1k.** The ImageNet-1k dataset (Deng et al., 2009) consists of 1,281,167 training images and 50,000 validation images, distributed across 1,000 distinct classes. The images in ImageNet-1K are normalized to have zero mean and unit variance. During training, the images are randomly resized and cropped to dimensions of 224x224 pixels, followed by horizontal flipping. For validation, the images are first resized to 256x256 pixels and subsequently center-cropped to 224x224 pixels. Similar to the methodology applied to the CIFAR datasets, the images are converted into temporal sequences using direct encoding (Rathi & Roy, 2021; Fang et al., 2021a). For performance comparison, the SEW ResNet (Fang et al., 2021a) architecture is used as the backbone model.

**CIFAR10-DVS.** The CIFAR10-DVS dataset (Li et al., 2017) is a neuromorphic dataset derived from CIFAR-10 through conversion using a Dynamic Vision Sensor (DVS) camera. It comprises 10,000 event-based images with an expanded resolution of 128×128 pixels. The integration of events into frames is performed using the SpikingJelly framework (Fang et al., 2023). No data augmentation or TEBN techniques (Duan et al., 2022) are applied to the CIFAR10-DVS dataset. For performance comparison, the Spiking-VGG11 (Huang et al., 2024) architecture (referred to as VGG11) is adopted as the backbone model.

**DVS-Gesture.** The DVS-Gesture dataset (Amir et al., 2017) is a neuromorphic dataset that captures 11 distinct gestures performed by 29 participants under three different lighting conditions. It contains a total of 1,342 samples, with an average duration of 6.5 seconds per sample. The dataset is divided into a training set with 1,208 samples and a test set with 134 samples. Following prior work (Fang et al., 2021b), the events are integrated into frames using the SpikingJelly framework (Fang et al., 2023). No data augmentation techniques are applied to the DVS-Gesture dataset, and the Spiking-VGG11 (Huang et al., 2024) (VGG11) architecture is employed as the backbone for performance evaluation.

## D.2 TRAINING SETUP

**Training Details.** Table 6 lists the key hyperparameters and configurations required for training on the static datasets (CIFAR-10, CIFAR-100, ImageNet-1k), and neuromorphic datasets including CIFAR10-DVS and DVS-Gesture. Our experiments on CIFAR-10, CIFAR-100, CIFAR10-DVS, and DVS-Gesture datasets are conducted using NVIDIA GeForce RTX 3090 GPUs, each equipped with 24 GB of memory. The training process on ImageNet-1k is executed on eight NVIDIA RTX A6000 GPUs, each equipped with 48 GB of memory.

Table 6: Training hyperparameters and configurations.

|  | CIFAR-10 | CIFAR-100 | ImageNet-1k | CIFAR10-DVS | DVS-Gesture |
|---|---|---|---|---|---|
| Optimizer | SGD | SGD | AdamW | SGD | SGD |
| Epoch | 200 | 200 | 200 | 300 | 300 |
| Learning rate | 1e-1 | 1e-1 | 5e-2 | 5e-2 | 5e-2 |
| Batch size | 128 | 128 | 256 | 128 | 16 |
| Weight decay | 5e-5 | 5e-4 | 5e-4 | 5e-4 | 5e-4 |
| Momentum | 0.9 | 0.9 | - | 0.9 | 0.9 |
| Lr schedule | Cosine | Cosine | Cosine | Cosine | Cosine |
| Loss function | Cross-entropy | Cross-entropy | Cross-entropy | TET | TET |

# E    MORE RESULTS

## E.1    ABLATION STUDY

To investigate the impact of the proposed components on the model's performance, we conducted several ablation experiments on the CIFAR-10 dataset using ResNet18 as the backbone architecture. To exclude the effects introduced by multi-bit spikes of the I-LIF neuron, we set $T = 1$ for the vanilla ResNet18 and $D = 1$ for our model. As reported in Table 7, integrating PIT into the model significantly improves its performance, with accuracy increasing from 86.73% to 91.34%, marking a notable enhancement of 4.61%. Furthermore, by leveraging our proposed input distribution-aware initialization strategy and gradient correction term, the model's performance is further enhanced. These results demonstrate that, under ultra-low latency conditions ($T = 1, D = 1$), the introduction of PIT significantly enhances the performance of SNNs. Furthermore, these performance gains are not due to the use of multi-bit spikes but arise directly from the PIT itself.

Table 7: Comparison of different configurations of our proposed PIT using ResNet18 as backbone on CIFAR-10.

| Configurations | Learnable PIT | Input-Distribution-Aware Init | Gradient Rectifying | Accuracy |
|---|---|---|---|---|
| ResNet18 | ✗ | ✗ | ✗ | 86.73% |
| (i) | ✓ | ✗ | ✗ | 91.34% |
| (ii) | ✓ | ✓ | ✗ | 92.53% |
| Ours | ✓ | ✓ | ✓ | 93.23% |

Table 8: Comparison of different configurations of LIF, I-LIF, and our proposed PIT on CIFAR-10.

| Model | Neuron Model | Learnable PIT | Multi-bit Spikes | Params | $T \times D$ | Accuracy |
|---|---|---|---|---|---|---|
| ResNet18 | LIF | ✗ | ✗ | 11.18M | $4 \times 1$ | 94.22% |
| ResNet18 | I-LIF | ✗ | ✓ | 11.18M | $1 \times 4$ | 94.83% |
| ResNet18 | I-LIF | ✓ | ✗ | 11.18M | $4 \times 1$ | 95.34% |
| ResNet18 | I-LIF | ✓ | ✓ | 11.18M | $1 \times 4$ | 95.86% |

To compare the performance of our proposed PIT with LIF and I-LIF (Luo et al., 2024) under the same inference latency, we conducted an additional ablation study using ResNet18 as the backbone model on CIFAR10. The comparison results, as shown in Table 8, demonstrate that our proposed PIT achieves a 1.12% improvement in accuracy compared to vanilla ResNet18 (LIF). Furthermore, combining PIT with multi-bit spikes provides performance gains of 1.64% and 1.03% compared to vanilla ResNet18 (LIF) and ResNet18 (I-LIF), respectively, further validating the effectiveness of our proposed method.

To investigate the trade-off between temporal length ($T$) and firing levels ($D$), we evaluate different $T \times D$ configurations on CIFAR10 and CIFAR100 using ResNet18 as the backbone model. Table 9 demonstrates that configurations with fewer timesteps and more firing levels consistently outperform their counterparts. Specifically, the $1 \times 4$ configuration achieves the highest accuracy on both datasets (95.86% and 78.83%), surpassing the $4 \times 1$ configuration by 1.03% and 1.92%, respectively. This trend indicates that for static image datasets without inherent temporal structure, the multi-level firing mechanism contributes more significantly to model performance than temporal length ($T$).

Table 9: Impact of $T \times D$ configurations on SNNs' performance.

| Dataset | Architecture | Method | Params | $T \times D$ | Accuracy |
|---|---|---|---|---|---|
| | ResNet18 | PIT | 11.18M | $4 \times 1$ | 94.83% |
| CIFAR10 | ResNet18 | PIT | 11.18M | $2 \times 2$ | 95.47% |
| | ResNet18 | PIT | 11.18M | $1 \times 4$ | 95.86% |
| | ResNet18 | PIT | 11.18M | $4 \times 1$ | 76.91% |
| CIFAR100 | ResNet18 | PIT | 11.18M | $2 \times 2$ | 77.65% |
| | ResNet18 | PIT | 11.18M | $1 \times 4$ | 78.83% |

## E.2 Comparison of Energy Consumption

In this section, we provide a detailed comparison of the energy cost between LIF, I-LIF (Luo et al., 2024; Yao et al., 2025), and our spiking neuron models based on Synaptic Operations (SynOps) and Neuron Operations (NeuOps) incurred during neuron updates. SynOps consist of Accumulate (AC) and Multiply-And-Accumulate (MAC) operations. Following previous studies (Han et al., 2015; Horowitz, 2014), we assume that the operations are performed using 32-bit floating-point arithmetic under 45 nm CMOS technology, where an AC operation consumes 0.9 pJ and a MAC operation consumes 4.6 pJ. We conduct an extended analysis of energy cost using ResNet18 as the backbone model for the CIFAR10 dataset.

Table 10: Comparison of the energy cost for LIF, I-LIF, and our PIT using ResNet18 as the backbone model on CIFAR10.

| Model | Input Resolution | $T \times D$ | ACs (M) | MACs (M) | Params (M) | Energy ($\mu$J) |
|---|---|---|---|---|---|---|
| ResNet18 (LIF) | $32 \times 32 \times 3$ | $4 \times 1$ | 68.04 | 2.33 | 11.18 | 71.94 |
| ResNet18 (I-LIF) | $32 \times 32 \times 3$ | $1 \times 4$ | 56.67 | 2.33 | 11.18 | 61.71 |
| ResNet18 (PIT) | $32 \times 32 \times 3$ | $1 \times 4$ | 49.30 | 2.88 | 11.18 | 57.64 |

The results reported in the Table 10 demonstrate that our model (the last row) achieves lower energy consumption. The energy efficiency of our model can be attributed to the PIT we introduced, which dynamically adjusts the membrane potential distribution before firing and significantly reduces the firing rate, as verified in Figure 6. Collectively, although our method introduces a small number of additional MAC operations, our design minimizes overall energy consumption by reducing the firing rates of spiking neurons.

## E.3 Experimental Results on CIFAR10-DVS

Table 11 reports the experimental results on the CIFAR10-DVS dataset. Compared to previous approaches that focus on adjusting the membrane potential distribution before firing, such as MPBN (Guo et al., 2023b) and IM-Loss (Guo et al., 2022a), our method achieves superior performance with a lower-parameter network architecture (VGG11) under the same equivalent timesteps. Furthermore, compared to prior methods like Real Spike (Guo et al., 2022c) and Trainable Ternary Spike (Guo et al., 2024), integrating our method into VGG11 ($T = 10, D = 1$) achieves improvements of 5.92% and 3.92%, respectively.

Table 11: Comparison with SOTA methods on CIFAR10-DVS. Our results are reported as averages over three experimental runs with different random seeds.

| Method | Type | Architecture | Params | $T \times D$ | Accuracy |
|---|---|---|---|---|---|
| STBP-tdBN (Zheng et al., 2021) | SNN training | ResNet19 | 12.70M | $10 \times 1$ | 67.80% |
| PLIF (Fang et al., 2021b) | SNN training | PLIF Net | 37.31M | $20 \times 1$ | 74.80% |
| MPBN (Guo et al., 2023b) | SNN training | ResNet19 | 12.70M | $10 \times 1$ | 74.40% |
| IM-Loss (Guo et al., 2022a) | SNN training | ResNet19 | 12.70M | $10 \times 1$ | 72.60% |
| Dspike (Li et al., 2021) | SNN training | ResNet18 | 11.18M | $10 \times 1$ | 75.40% |
| TET (Deng et al., 2022) | SNN training | VGG11 | 9.33M | $10 \times 1$ | 77.33% |
| Trainable Ternary Spike (Guo et al., 2024) | SNN training | ResNet19 | 12.70M | $10 \times 2$ | 79.80% |
| | | VGG11 | 9.33M | $10 \times 2$ | 76.60% |
| Real Spike (Guo et al., 2022c) | SNN training | ResNet19 | 12.70M | $10 \times 1$ | 72.85% |
| | | VGG16 | 14.72M | $10 \times 1$ | 74.58% |
| **PIT (Ours)** | SNN training | VGG11 | 9.33M | $10 \times 1$ | **80.50%** $\pm$ 0.08 |
| | | VGG11 | 9.33M | $10 \times 4$ | **81.50%** $\pm$ 0.13 |

## E.4 Experimental Results on DVS-Gesture

To further evaluate the effectiveness of our method on neuromorphic datasets, we conducted experiments on the DVS-Gesture dataset. The preprocessing steps followed the procedure outlined in prior work (Fang et al., 2021b), with the data interpolated to $T = 10$ using the SpikingJelly framework (Fang et al., 2023). VGG11 was adopted as the backbone, consistent with previous work (Huang et al., 2024). As shown in Table 12, it is evident that our method achieves state-of-the-art classification

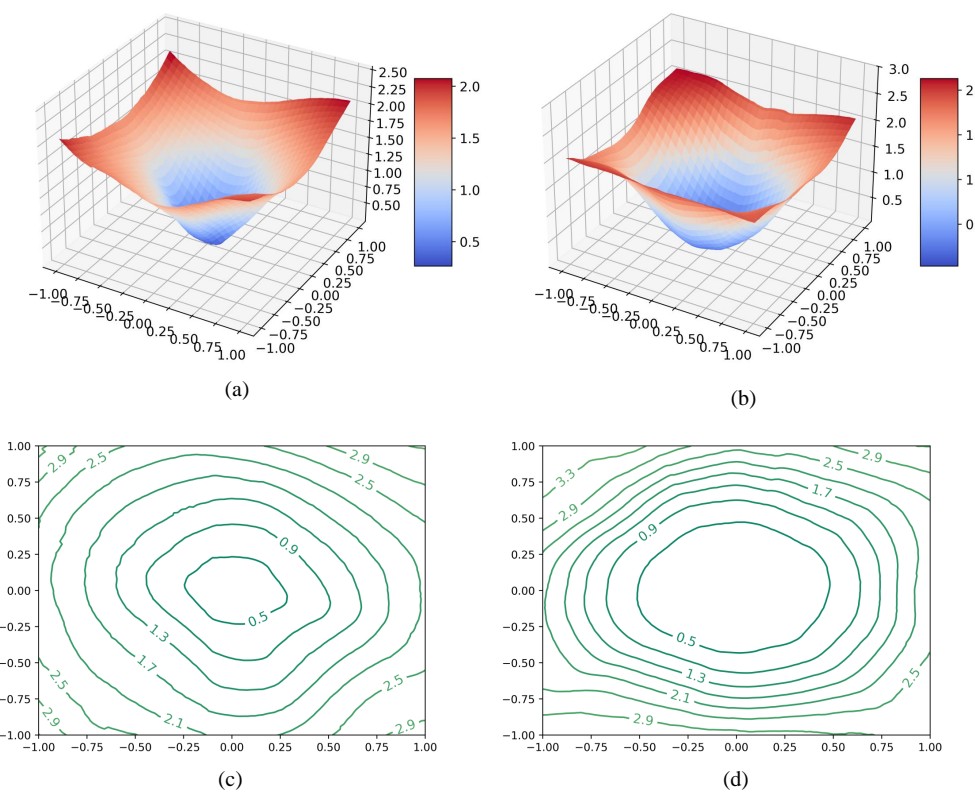

Figure 5: Comparison of the loss landscape of (a, c) vanilla ResNet18 and (b, d) ResNet18 integrated with PIT models in terms of 3D surface and 2D contour plots on CIFAR10.

accuracy under the same network architecture. Specifically, our method, i.e., VGG11 integrated with PIT ($T = 20, D = 1$), surpasses Real Spike (Guo et al., 2022c) by 2.02%. Moreover, VGG11 integrated with PIT ($T = 20, D = 2$) outperforms Trainable Ternary Spike (Guo et al., 2024) by 1.84%, demonstrating the effectiveness of our approach under equivalent inference timesteps.

Table 12: Comparison with SOTA methods on DVS-Gesture. Our results are reported as averages over three experimental runs with different random seeds.

| Method | Type | Architecture | Params | $T \times D$ | Accuracy |
|---|---|---|---|---|---|
| PLIF (Fang et al., 2021b) | SNN training | PLIF Net | 37.31M | $20 \times 1$ | 97.57% |
| CLIF (Huang et al., 2024) | SNN training | VGG11 | 9.33M | $20 \times 1$ | 97.92% |
| Trainable Ternary Spike (Guo et al., 2024) | SNN training | VGG11 | 9.33M | $20 \times 2$ | 96.42% |
| Real Spike (Guo et al., 2022c) | SNN training | VGG11 | 9.33M | $20 \times 1$ | 96.21% |
| **PIT (Ours)** | SNN training | VGG11 | 9.33M | $20 \times 1$ | **98.23%** $\pm\, 0.05$ |
| | | VGG11 | 9.33M | $20 \times 2$ | **98.26%** $\pm\, 0.08$ |

## E.5 EXTENDED VISUALIZATION OF LOSS LANDSCAPE

To evaluate the impact of the proposed PIT on the optimization procedure, we conducted a comparative analysis of the loss landscape of our model against its vanilla counterpart. ResNet18 was employed as the backbone architecture, with $T = 1$ and $D = 1$ set to exclude any effects introduced by the multi-bit spikes of the I-LIF neuron. By utilizing the technique introduced by Li et al. (2018), we plot the 3D surface and 2D contour of the loss landscape of models in Figure 5. From Figure 5, we could observe that the model integrated with PIT exhibits a markedly smoother loss landscape around the local minima, which facilitates more stable and faster convergence during training.

### E.6  EXTENDED VISUALIZATION OF FIRING RATES STATISTICS

To analyze the impact of our method on firing rates, we track the average firing rates of the ResNet18 integrated with PIT and its vanilla counterpart across each layer over time, as shown in Figure 6. Specifically, the ResNet18 integrated with PIT exhibits lower average firing rates (purple line) and demonstrates a descending trend with respect to the time step. In contrast, the vanilla ResNet18 tends to fire more frequently in later time steps and shows higher firing rates, particularly in the earlier layers. This phenomenon can be attributed to the ability of our input-distribution-aware PIT to dynamically adjust the potential distribution before the firing operation, which is absent in vanilla spiking models.

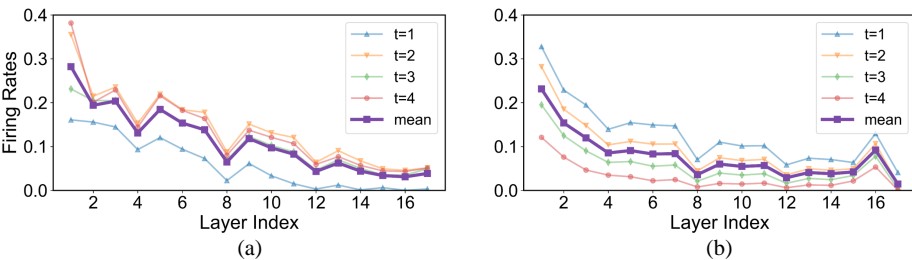

Figure 6: Comparison of the firing rates statistics for (a) vanilla ResNet18 and (b) the ResNet18 integrated with PIT models on CIFAR10.

### E.7  EXTENDED VISUALIZATION OF PIT DISTRIBUTION ACROSS LAYERS

To inspect the evolving dynamics of our proposed PIT during the training stage, we use ResNet18 ($T = 1, D = 4$) as the backbone and visualize the parameter distribution involved in PIT across layers at the beginning of training and after convergence on CIFAR-10 and CIFAR-100 datasets, as illustrated in Figure 7. It can be observed that, at the beginning of training, the parameter distribution of PIT is relatively concentrated. After convergence, the parameter distribution of PIT exhibits significant inter-layer differences, demonstrating that PIT can dynamically learn and adjust according to the features at different levels.

## F  DISCUSSION ON THE STRUCTURE OF PIT

The conceptualization of PIT draws inspiration from the adaptive learning mechanisms inherent in biological systems, particularly the mechanisms underlying neural plasticity and the modulation of membrane potentials (Salaj et al., 2021). As evidenced by previous neuroscience findings, neurons exhibit a remarkable ability to modify their neurophysiological characteristics in response to dynamically changing environmental stimuli. This plasticity plays a critical role in enabling efficient information encoding and processing, as it allows neurons to flexibly adjust their neuronal activities and optimize their responses to external inputs.

In this work, to conserve parameters and memory, the transformation matrix of PIT is designed in a diagonal form. However, in principle, it can adopt a more general structure. Specifically, if the off-diagonal elements of the transformation matrix are non-zero, it indicates that a neuron receives modulation signals (the sub-threshold voltages) of neighboring neurons before firing. This design mimics the excitatory-inhibitory (E-I) balance observed in biological neural groups (Zhou & Yu, 2018). Such a sub-threshold modulation mechanism could be effectively simulated by our proposed PIT method. Additionally, neurons in biological systems may exhibit bursting behavior when responding to complex stimuli, which can also be represented by applying our proposed PIT after the firing operation. The design of integrating PIT into neuronal dynamics in a conjugate manner encapsulates both adaptive regulation and efficient neural coding.

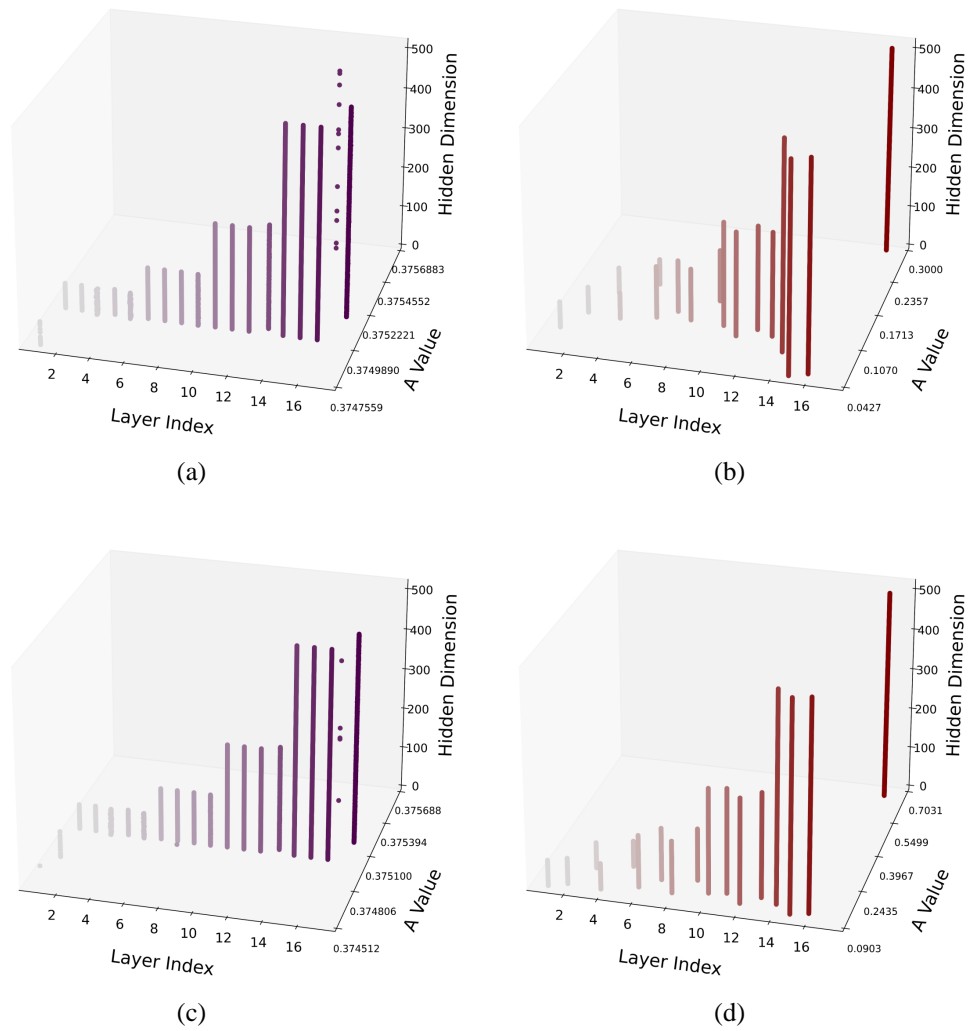

Figure 7: Comparison of the parameter distribution in PIT across the layers of the ResNet18 integrated with the PIT model at the first epoch (a) and the last epoch (b) on CIFAR10. Comparison of the parameter distribution in PIT across the layers of the ResNet18 integrated with the PIT model at the first epoch (c) and the last epoch (d) on CIFAR100.

# G   THE USE OF LARGE LANGUAGE MODELS

In this paper, Large Language Models (LLMs) are only used for polishing and correcting grammatical errors. Their usage does not affect the core methodology, scientific rigor, or originality of this research.

