# OpenReview forum: "Advancing Spatiotemporal Representations in Spiking Neural Networks via Parametric Invertible Transformation"
_ICLR.cc/2026/Conference — ICLR 2026 Poster_

### Official Review · Reviewer_KE4i · 2025-10-21

**Soundness:** 2
**Presentation:** 3
**Contribution:** 2
**Rating:** 4
**Confidence:** 5

**Summary:**

The paper builds on I-LIF–style neurons and instantiates firing as uniform integer quantization of membrane potential with a learnable per-channel step. Concretely, it applies a conjugate scaling scheme ($A^{-1}$ before rounding and $A$ after) so that $s_t^l=A_t^l,\mathrm{clip}(\lfloor (A_t^l)^{-1}u_t^l\rceil,0,D)$ and $v_t^l=u_t^l-s_t^l$, alongside the standard charging equation. The scale $A_t^l$ is parameterized per channel, initialized by a channel-wise $3\sigma$ rule, and trained end-to-end with a rectified surrogate-gradient (STE-like) update. For $T=1$ and small $D$ (e.g., $D=4$), the neuron reduces to a one-shot multi-level activation quantizer. Inference remains event-driven by folding $A$ into downstream weights. Experiments train the SNN directly rather than via ANN→SNN conversion, and the paper positions the approach as an engineering refinement that reconciles SNN charge–discharge–reset semantics with practical quantization.

**Strengths:**

The paper presents a coherent synthesis: framing firing as uniform integer quantization wrapped by conjugate scaling, with per-channel learnable steps and a pragmatic $3\sigma$ initializer, plus a rectified surrogate gradient near quantization boundaries. In terms of quality, the components are principled and practical—easy to implement in standard toolchains, trainable end-to-end, and compatible with event-driven deployment.

**Weaknesses:**

1. With temporal dynamics effectively disabled (e.g., $T=1$ so no carryover via $v_t$), Eqs. (9–11) collapse to a static per-layer mapping
$a^{(l)} = Q_{A^{(l)},D}\big(W^{(l)} a^{(l-1)}\big)$,
where $Q_{A,D}(z)=A\,\mathrm{clip}(\mathrm{round}(z/A),0,D)$.
This is a per-channel uniform activation quantizer with a learnable step, i.e., a QAT-style layer; the model no longer exhibits SNN temporal dynamics. When $T\times D = 1\times 4$, this is exactly the one-shot 5-level quantizer $Q_{A,4}(z)=A,\mathrm{clip}(\mathrm{round}(z/A),0,4)$; with $T=1$ the temporal state is negligible, so the layer behaves like a QAT activation rather than a classical LIF gate. The paper does not clearly acknowledge this equivalence and risks overstating novelty relative to standard QAT (per-channel scale learning with STE).

2. Missing baselines and fairness. For $T=1$ and $D=2/4$ in the experiments, the correct baseline is an ANN with uniform QAT using the same level count and per-channel scale learning (same rounding rule, same folding of scales). These head-to-head comparisons are missing, so it is unclear whether the gains come from SNN dynamics or simply from quantization.

3. Temporal dynamics conflation. Although the scheme preserves charge–discharge–soft-reset, firing is no longer a spike-time thresholding mechanism but a multi-level amplitude code. Claims attributed to SNN dynamics should be isolated by sweeping $T$ and $\lambda$. With $T=1$, Eq. (9) contributes little temporal effect; with $T>1$, residual carryover arises from $v_t = u_t - s_t$. Accuracy results should be reported.

**Questions:**

1. Explicit equivalence: acknowledge that Eq. (10) is a uniform quantizer and provide the mapping $s_t^l = Q_{A_t^l,D}(u_t^l)$ with $Q_{A,D}(z)=A,\mathrm{clip}(\mathrm{round}(z/A),0,D)$. Clarify what is fundamentally new beyond scale learning plus STE.

2. QAT baselines: add ANN+QAT baselines with identical level counts (e.g., 5 levels for $D=4$), per-channel scale learning, and the same rounding and folding; compare accuracy.

3. Temporal isolation: sweep $T$ and report accuracy.

4. Energy evidence and fair accounting. Since this reduces to quantization (e.g., $T\times D=1\times 2/4$), report experiments against an ANN+QAT baseline with the same levels/scales and matched energy budgets.

---

Shen et al., “Are Conventional SNNs Really Efficient? A Perspective from Network Quantization,” CVPR 2024; https://arxiv.org/pdf/2311.10802  ￼

---

> ### Author Response · Authors · 2025-11-22
>
> Thank you for your precious time and insightful comments. We address each concern below.
>
> __Reply to Weakness 1:__ We sincerely thank the reviewer for this insightful observation.
> In fact, the longstanding practice in the SNN community of validating methods on static image datasets without temporal structure needs reconsideration. The temporal dynamics of SNNs, which are distinct from those of QNNs and ANNs, represent a critical aspect that requires thorough examination and discussion. For static image datasets, our method degenerates into a QAT-like approach combined with our proposed rectified surrogate gradient function (Eq. (13)). However, we want to clarify that the PIT design proposed in this paper is time-varying. This design enables the use of a heterogeneous quantization strategy for sequential learning, which is absent in standard QAT approaches. Moreover, this time-varying design results in adaptive firing patterns and delivers performance gains. Specifically:
>
> - **The design of PIT can adapt to the structure underlying both static and neuromorphic data.**
> In this work, we clarify that one of the core purposes of PIT is to enable SNNs to adapt to input data patterns and distributions, thereby enhancing their temporal processing and modeling capabilities, which fundamentally differs from standard QAT approaches.
> For static image datasets (e.g., CIFAR100, ImageNet), the data inherently lacks temporal structure and requires converting static images into sequences, typically by replication along the time axis. Therefore, it is reasonable to directly set $T=1$ for static image datasets for training efficency, as commonly adopted in prior work [1,2].
> However, for neuromorphic datasets (CIFAR10-DVS, DVS-Gesture in Tables 7-8), our proposed PIT is inherently time-varying, which differs substantially from QAT. Specifically, $A_t^l$ is jointly determined by the residual membrane potential $v_{t-1}^l$, decay factor $\lambda$, and input signal at the current timestep (Eq. 10-12). This recurrent dependency indicates that PIT performs non-uniform temporal quantization to modulate firing thresholds and compensate outputs simultaneously, which possess a temporal adaptive characteristic absent in standard QAT works.
>
> - **Time-varying dynamics and temporal adaptation.**
> PIT provides input-distribution-aware, temporally-adaptive firing that better exploits the temporal characteristics of event-based data. This time-varying design enables SNNs to exhibit different firing patterns and dynamics across timesteps, relying on the inherent recurrent dependency in SNNs and input stream distribution (Figure 4c). If we simply applied QAT methods, the model would operate under homogeneous quantization functions along the sequence, losing the adaptive capacity for temporal processing.
> This is evidenced by our superior performance on neuromorphic datasets: +5.92% over Real Spike on CIFAR10-DVS (Table 11) and +2.02% on DVS-Gesture (Table 12).
>
> - **Balancing firing sparsity and performance.**
> Balancing SNN the computational overhead associate with firing density with performance is an important challenge faced by current community. Our method achieves a better trade-off between energy cost and model performance by decoupling the adaptive modulation mechanism in the temporal dimension from multi-bit firing behavior in the spatial dimension, in an input-distribution-aware manner. Table 10 shows that our method achieves lower energy consumption (57.64 µJ vs. 61.71 µJ) while not compromising model performance.
>
> Collectively, while PIT at $T=1$ shares some similarities with QAT-style quantization, it maintains fundamentally different temporal dynamics for event-based data processing through time-varying transformations, a characteristic essential for SNNs but absent in standard QAT approaches that do not exhibit time-dependent dynamics and sequential modeling capability.
>
> [1] Luo, X., et al. Integer-valued training and spike-driven inference spiking neural network for high-performance and energy-efficient object detection. ECCV 2024.
>
> [2] Yao, M., et al. Scaling spike-driven transformer with efficient spike firing approximation training. IEEE TPAMI, 2025.

---

> > ### Author Response · Authors · 2025-11-22
> >
> > __Reply to Weakness 2:__ We thank the reviewer for this valuable suggestion. We conducted additional experiments comparing our method with an ANN using uniform QAT with the same level count and per-channel scale learning on CIFAR-10, CIFAR-100, and ImageNet-1k datasets. The results are summarized below:
> >
> > | Dataset      | Method | Architecture | T×D | Accuracy |
> > |--------------|--------|--------------|-----|----------|
> > | CIFAR-10     | QAT    | ResNet-18    | N/A | 95.14%   |
> > |      |    | ResNet-19    | N/A | 96.05%   |
> > |    | PIT    | ResNet-18    | 1×4 | 95.86%   |
> > |     |   | ResNet-19    | 1×4 | 96.72%   |
> > | CIFAR-100    | QAT    | ResNet-18    | N/A | 77.91%   |
> > |     |    | ResNet-19    | N/A | 80.87%   |
> > |      | PIT    | ResNet-18    | 1×4 | 78.83%   |
> > |     |    | ResNet-19    | 1×4 | 81.59%   |
> > | ImageNet-1k  | QAT    | ResNet-18    | N/A | 68.46%   |
> > |    |    | ResNet-34    | N/A | 71.54%   |
> > |   | PIT    | ResNet-18    | 1×4 | 69.39%   |
> > |     |    | ResNet-34    | 1×4 | 72.66%   |
> >
> > Our method demonstrates consistent performance improvements across all three static datasets. Notably, on ImageNet-1k with ResNet-34, our approach achieves 1.12% higher accuracy compared to the QAT baseline. These improvements stem from our rectified surrogate gradient function (Eq. 13), which facilitates the gradient-based optimization process to find solutions with better generalizability.
> >
> > In addition, we conducted comparative experiments on event-based datasets (CIFAR10-DVS and DVS-Gesture), and the results are as follows:
> >
> > | Dataset      | Method | Architecture | T×D  | Accuracy |
> > |--------------|--------|--------------|------|----------|
> > | CIFAR10-DVS  | QAT    | VGG-11       | N/A  | 78.30%   |
> > |       | PIT    | VGG-11       | 10×1 | 80.50%   |
> > | DVS-Gesture  | QAT    | VGG-11       | N/A  | 96.97%   |
> > |     | PIT    | VGG-11       | 20×1 | 98.23%   |
> >
> > On event-based datasets, the QAT method using the same level count and per-channel scale learning shows notably inferior performance, achieving 2.2% and 1.26% lower accuracy compared with our PIT on CIFAR10-DVS and DVS-Gesture, respectively. The superior performance of our method is attributed to two key factors: (1) the rectified surrogate gradient function design, and (2) our input-distribution-aware strategy for PIT, which enables time-varying adaptation to the input stream rather than applying a uniform quantization scheme along the temporal dimension as in QAT methods.
> >
> > These comparison results demonstrate that the gains come from both the SNN dynamics and our specific designs. While static datasets show moderate improvements (as the data lack inherent temporal dependency), the substantial performance gap on event-based datasets clearly highlights the advantages of our method in leveraging temporal dynamics, which is absent in standard QAT approaches that treat temporal information uniformly.
> >
> >
> > __Reply to Weakness 3:__ We thank the reviewer for this important point regarding the contribution of temporal dynamics. To investigate the impact of different T×D combinations on model performance, we conducted additional ablation studies on CIFAR-10, CIFAR-100, and DVS-Gesture datasets using ResNet-18 and VGG-11 architectures. The results are summarized below:
> >
> > | Dataset   | Architecture | Method | T×D | Accuracy |
> > |-----------|--------------|--------|-----|----------|
> > | CIFAR-10  | ResNet-18    | PIT    | 4×1 | 94.83%   |
> > |     | ResNet-18    | PIT | 2×2 | 95.47%   |
> > |    | ResNet-18    | PIT | 1×4 | 95.86%   |
> > | CIFAR-100 | ResNet-18    | PIT    | 4×1 | 76.91%   |
> > |   | ResNet-18    | PIT  | 2×2 | 77.65%   |
> > |   | ResNet-18    | PIT  | 1×4 | 78.83%   |
> >
> > The results reveal that for static image data lacking inherent temporal dependencies, increasing total timesteps T does not provide benefits and actually leads to slight performance degradation. This confirms the reviewer's observation that with T=1, temporal dynamics contribute minimally to static tasks, and our gains primarily come from the multi-level firing mechanism and rectified gradient design.
> >
> > For event-based dataset (DVS-Gesture), the results are as follows:
> >
> > | Architecture | Method | T×D  | Accuracy |
> > |--------------|--------|------|----------|
> > | VGG-11       | PIT    | 10×2 | 96.87%   |
> > | VGG-11       | PIT    | 20×1 | 98.23%   |
> > | VGG-11       | PIT    | 10×4 | 97.45%   |
> > | VGG-11       | PIT    | 20×2 | 98.26%   |
> >
> > In contrast, for event-based data with inherent temporal structure, increasing total timesteps T yields additional performance gains. Concretely, we observe that longer temporal integration (higher T) improves model's accuracy. This demonstrates that our time-varying design of PIT becomes more effective when processing sequential data with temporal dependencies.
> >
> > This clarifies that our method adapts to task characteristics: exploiting multi-bit coding efficiency for static data while leveraging the input-distribution-aware strategy of PIT for event-based data with temporal structure.

---

> ### Author Response · Authors · 2025-11-22
>
> __Q1:__
>
> > Explicit equivalence: acknowledge that Eq. (10) is a uniform quantizer and provide the mapping $s_t^l = Q_{A_t^l,D}(u_t^l)$ with $Q_{A,D}(z)=A,\mathrm{clip}(\mathrm{round}(z/A),0,D)$. Clarify what is fundamentally new beyond scale learning plus STE.
>
> __Reply:__ We thank the reviewer for this question.
> In fact, as we analyzed in Section 3.2, due to the inherent temporal dynamics in SNNs, the introduced PIT is designed in a time-varying form based on the input-stream distribution.
> For static image datasets, we agree that Eq. (10) can be expressed as a uniform quantizer: $s_t^l = Q_{A_t^l,D}(u_t^l)$, where $Q_{A,D}(z) = A \cdot \text{clip}(\text{round}(z/A), 0, D)$. This formulation resembles QAT, and we optimize it using our proposed rectified surrogate gradient function to facilitate learning. However, our method differs the QAT method significantly in terms of firing patterns and adaptation to event-based data for sequential modeling tasks.
> Specifically, our method introduces fundamentally new contributions beyond scale learning and STE in the following aspects:
>
> 1. **Time-varying, input-distribution-aware design:** Unlike standard QAT with fixed per-channel scales, our $A_t^l$ is time-varying and jointly determined by residual membrane potential $v_{t-1}^l$, decay factor $\lambda$, and input signal at the current timestep (Eq. 10-12). This recurrent dependency enables non-uniform temporal quantization that adapts to the input stream, which is a characteristic absent in QAT methods that apply homogeneous quantization along temporal dimensions.
>
> 2. **Rectified surrogate gradient function:** We propose a specialized gradient estimator that addresses the gradient mismatch between integer-valued forward propagation and continuous backward propagation, with penalty terms near rounding boundaries. This differs from standard STE and demonstrates superior optimization performance (see our response to **Weakness 2**: +1.12% on ImageNet-1k vs. QAT baseline).
>
> 3. **Temporal dynamics exploitation:** For event-based data with inherent temporal structure (CIFAR10-DVS, DVS-Gesture in Tables 7-8), our time-varying PIT leverages SNN's recurrent dynamics to achieve adaptive firing patterns across timesteps. This temporal adaptation capability is fundamentally different from QAT's static quantization and yields significant performance gains: +2.2% over QAT baseline on CIFAR10-DVS and +1.26% on DVS-Gesture.
>
> We have added explicit clarifications on these distinctions in the revised manuscript.
>
>
>
> __Q2:__
>
> > QAT baselines: add ANN+QAT baselines with identical level counts (e.g., 5 levels for $D=4$), per-channel scale learning, and the same rounding and folding; compare accuracy.
>
> __Reply:__ We thank the reviewer for this suggestion. Please refer to our comprehensive response to **Weakness 2**, where we provide detailed comparisons with the ANN+QAT baseline using identical level counts, per-channel scale learning, and the same rounding and folding mechanisms across CIFAR-10, CIFAR-100, ImageNet-1k, CIFAR10-DVS, and DVS-Gesture datasets. The results demonstrate consistent performance improvements of our method, with particularly notable gains on event-based datasets (+2.2% on CIFAR10-DVS, +1.26% on DVS-Gesture) attributed to our time-varying, input-distribution-aware PIT design and rectified surrogate gradient function.
>
>
> __Q3:__
>
> > Temporal isolation: sweep $T$ and report accuracy.
>
> __Reply:__ Please refer to our comprehensive response to **Weakness 3**, where we provide detailed ablation studies sweeping T across different T×D combinations on CIFAR-10, CIFAR-100, and DVS-Gesture datasets. The results demonstrate that: (1) for static image datasets, increasing $T$ provides no benefits as static images lack inherent temporal structure, and (2) for event-based datasets with temporal dependencies, increasing $T$ yields substantial performance gains, confirming the effectiveness of our time-varying and input-distribution-aware design of PIT when processing sequential data.

---

> ### Author Response · Authors · 2025-11-22
>
> __Q4:__
>
> > Energy evidence and fair accounting. Since this reduces to quantization (e.g., $T\times D=1\times 2/4$), report experiments against an ANN+QAT baseline with the same levels/scales and matched energy budgets.
>
> __Reply:__ We thank the reviewer for this question. Following prior work [3], we adopt the Synaptic Arithmetic Computation Effort (S-ACE) and Neuromorphic Synaptic Arithmetic Computation Effort (NS-ACE) metrics to provide fair energy comparisons. We conduct additional experiments on CIFAR-10 and CIFAR-100, where all compared methods assume computation under 16-bit weights. Our method is compared against both vanilla SNNs and the QAT baseline with the same level count and per-channel scale learning:
>
> | Dataset   | Architecture | Method | T×D | S-ACE (G) | NS-ACE (G) | Params (M) | Accuracy |
> |-----------|--------------|--------|-----|-----------|------------|------------|----------|
> | CIFAR-10  | ResNet-18    | LIF    | 4×1 | 35.84     | 6.45       | 11.18      | 94.22%   |
> |           | ResNet-18    | QAT    | N/A | 35.84     | -          | 11.18      | 95.14%   |
> |           | ResNet-18    | PIT    | 1×4 | 35.84     | 4.31       | 11.18      | 95.86%   |
> | CIFAR-100 | ResNet-18    | LIF    | 4×1 | 35.84     | 7.88       | 11.18      | 73.25%   |
> |           | ResNet-18    | QAT    | N/A | 35.84     | -          | 11.18      | 77.91%   |
> |           | ResNet-18    | PIT    | 1×4 | 35.84     | 5.73       | 11.18      | 78.83%   |
>
> Our method achieves better accuracy than both LIF and QAT baselines under identical S-ACE budgets, which can be attributed to the rectified surrogate gradient design for learning our PIT.
> Compared to vanilla LIF, our method exhibits lower NS-ACE, which benefits from the adaptive modulation mechanism of PIT that results in reduced firing activity.
> These results provide evidence that our method delivers genuine improvements in both accuracy and energy efficiency under fair accounting conditions. We have included these comparisons in the revised manuscript.
>
> [3] Shen et al., "Are Conventional SNNs Really Efficient? A Perspective from Network Quantization," CVPR 2024.

---

### Official Review · Reviewer_rRf7 · 2025-10-25

**Soundness:** 3
**Presentation:** 3
**Contribution:** 3
**Rating:** 6
**Confidence:** 2

**Summary:**

This paper proposes a Parametric Invertible Transformation (PIT) and provides a theoretical analysis of the spatiotemporal representation space and capacity of Spiking Neural Networks (SNNs). Experimental results demonstrate improved accuracy on several datasets.

**Strengths:**

+ The paper is well written and clearly organized.

+ The proposed PIT effectively enhances information representation by transforming both membrane potentials and spikes in spiking neurons.

+ The introduction of an auxiliary gradient correction term further facilitates SNN optimization.

+ The paper provides comprehensive empirical comparisons with state-of-the-art models.

**Weaknesses:**

- While energy efficiency is presented as the main motivation of this work, the validation in this aspect appears rather limited. Simply calculating the computational energy is not sufficient; memory access cost should also be considered. A more detailed analysis or discussion on this point is recommended.

- Compared with E-SpikeFormer, the proposed method achieves only a 0.7% improvement under the same model size. It would be helpful to include a more in-depth comparison with E-SpikeFormer to better demonstrate the advantages and strengths of your approach.

**Questions:**

See in Weakness

---

> ### Author Response · Authors · 2025-11-22
>
> Thanks for your efforts in reviewing our paper and your recognition of the contribution and extensibility of our work. The response to your questions is given piece by piece as follows.
>
>
> __Weakness 1:__
>
> > While energy efficiency is presented as the main motivation of this work, the validation in this aspect appears rather limited. Simply calculating the computational energy is not sufficient; memory access cost should also be considered. A more detailed analysis or discussion on this point is recommended.
>
>
> __Reply:__ We thank the reviewer for this constructive suggestion. While our initial analysis focused on computational energy, we acknowledge that memory access cost is equally important for a comprehensive assessment. Following your recommendation, we have added memory footprint comparisons during training:
>
> | Dataset     | Architecture | Model | Params | T×D | Memory   | Accuracy |
> |-------------|--------------|-------|--------|-----|----------|----------|
> | CIFAR10     | ResNet18     | LIF   | 11.18M | 4×1 | 2.21 GB  | 94.22%   |
> |             | ResNet19     | LIF   | 12.70M | 4×1 | 4.45 GB  | 95.49%   |
> |             | ResNet18     | PIT   | 11.18M | 1×4 | 2.43 GB  | 95.86%   |
> |             | ResNet19     | PIT   | 12.70M | 1×4 | 4.70 GB  | 96.72%   |
> | ImageNet-1k | SEW ResNet18 | IF    | 11.69M | 4×1 | 13.65 GB | 63.18%   |
> |             | SEW ResNet34 | IF    | 21.79M | 4×1 | 18.86 GB | 67.04%   |
> |             | SEW ResNet18 | PIT   | 11.69M | 1×4 | 17.87 GB | 69.39%   |
> |             | SEW ResNet34 | PIT   | 21.79M | 1×4 | 24.66 GB | 72.66%   |
>
>
> On CIFAR10, our method incurs minimal additional memory overhead (+0.22 GB for ResNet-18, +0.25 GB for ResNet-19) while achieving substantially higher accuracy (+1.64% and +1.23%).
> While our method requires additional memory on ImageNet-1k, the performance gains are substantial (+6.21% and +5.62% respectively). This trade-off is favorable given that the accuracy improvements significantly outweigh the moderate memory cost increase.
> While training memory increases moderately, our method maintains advantages in inference energy efficiency through reduced firing activity (Figure 6).
> We have included this memory analysis and discussion in the revised manuscript to provide a more complete picture of the computational overhead and performance trade-offs.
>
>
>
> __Weakness 2:__
>
> > Compared with E-SpikeFormer, the proposed method achieves only a 0.7% improvement under the same model size. It would be helpful to include a more in-depth comparison with E-SpikeFormer to better demonstrate the advantages and strengths of your approach.
>
>
> __Reply:__ We thank the reviewer for this valuable suggestion. To provide more comparisons with E-SpikeFormer [1], we conducted additional experiments on ImageNet-1k using E-SpikeFormer architectures of different scales:
>
> | Method | Architecture    | Params | T×D | Accuracy |
> |--------|-----------------|--------|-----|----------|
> | SFA    | E-SpikeFormer-S | 5.11M  | 1×4 | 75.30%   |
> | SFA    | E-SpikeFormer-M | 10.02M | 1×4 | 78.50%   |
> | PIT    | E-SpikeFormer-S | 5.11M  | 1×4 | 76.00%   |
> | PIT    | E-SpikeFormer-M | 10.02M | 1×4 | 79.41%   |
>
>
> Results show that our method achieves performance gains over SFA on E-SpikeFormer-S and E-SpikeFormer-M, demonstrating consistent benefits across different model scales.
> The improvements are more modest compared to architectures like ResNet and Spike-driven Transformer [2] because E-SpikeFormer already employs integer-valued training through its SFA mechanism. Since both methods operate in the integer domain, the main source of our improvement comes from the rectified surrogate gradient function and the input-distribution-aware design of PIT, which provides smooth gradient flow during training and facilitates finding a solution with better generalizability.
>
> [1] Yao, M., et al. Scaling spike-driven transformer with efficient spike firing approximation training. IEEE TPAMI, 2025.
>
> [2] Yao, M., et al. Spike-driven transformer. Advances in neural information processing systems, 2023.

---

### Official Review · Reviewer_p6Qe · 2025-10-29

**Soundness:** 3
**Presentation:** 3
**Contribution:** 3
**Rating:** 4
**Confidence:** 4

**Summary:**

This paper introduces a innovative solution to the long-standing problem of limited representational power and training instability in Spiking Neural Networks (SNNs). Its core contribution is the Parametric Invertible Transformation (PIT), a conjugate transform applied before and after the neuron's firing process. The elegant design allows the model to learn with rich, ANN-like continuous representations during training, while a reparameterization trick recovers the efficient, event-driven spike-based computation at inference.

**Strengths:**

1. The novelty of the PIT concept. Instead of incrementally improving existing SNN components, it fundamentally rethinks how information flows and transforms within a neuron.
2. Empirical support. The experimental results provide evidence for the method's effectiveness and generality with SOTA results on a wide range of standard datasets.

**Weaknesses:**

1. Redefining the SNN Paradigm: By introducing continuous-valued transformations during training, the PIT method blurs the lines between traditional SNNs and ANNs with specialized quantization. This raises the question of whether we are still strictly in the SNN domain or creating a new class of SNN-inspired, highly efficient hybrid networks. The paper could benefit from a more explicit discussion of its philosophical positioning within the broader neural network landscape.
2. Unaddressed Trade-off between Temporal Dynamics and Bit-Depth. The paper relies on the T×D metric to claim comparable latency with prior work, assuming that trading temporal steps (T) for higher bit-depth (D) is an equitable exchange. This assumption is could be a weakness as it overlooks a critical trade-off. By decomposing a D-bit spike into D sequential sub-steps within a single logical timestep (as detailed in Appendix A), the model's architecture fundamentally shifts from a truly sequential processor to a static one with enhanced precision. A model with high T and low D (e.g., T=4, D=1) can integrate information across distinct and potentially distant time steps, making it suitable for tasks with genuine temporal dependencies. In contrast, the authors' preferred configuration of low T and high D (e.g., T=1, D=4) largely sacrifices this temporal processing capability in favor of higher representational richness within an isolated moment. The paper fails to acknowledge or discuss this fundamental architectural trade-off. This limits the claimed generality of the method, as its effectiveness on truly temporal tasks (like video analysis or complex time-series forecasting) compared to traditional high-T SNNs is not established and remains questionable. Furthermore, the equivalence of latency under the T×D metric is an idealization that may not hold on real hardware, where micro-looping overhead for high D could introduce different latency characteristics compared to processing distinct high-T steps.

**Questions:**

1. On the Fairness of the T×D Comparison: The paper uses T×D as a measure of equivalent inference latency, comparing their method (e.g., T=1, D=4) with prior works (e.g., T=4, D=1 or T=2, D=2).While T×D can be a proxy for total operations in some cases, different combinations of T and D can have different implications for actual hardware latency, memory access patterns, and the model's dynamic properties. For instance, a larger T implies longer sequential dependencies, while a larger D might require more complex computation within a single step. Could the authors discuss this trade-off in more detail? What are the limitations of this comparison metric?

2. On the Sensitivity of the Rectified Surrogate Gradient: The rectified surrogate gradient proposed in Eq. (13) is an interesting design that aims to reduce training oscillations by penalizing inputs near the rounding boundary. The computation introduces a hyperparameter λ, which is set to 0.01 in Appendix B. How sensitive is the model's performance to the choice of λ? A sensitivity analysis on the value of λ would be valuable for demonstrating the robustness and practical utility of this gradient correction term.

3. On the Distinction from Quantized Neural Networks (QNNs): In the low-latency regime of T=1 and D>1, the proposed model strongly resembles a Quantized Neural Network, where the integer spike value acts as a quantized activation and the neuron's dynamics effectively become a stateful quantization function. Could the authors elaborate on the fundamental distinctions between their approach (specifically in the T=1 case) and mainstream QNN research? While the inspiration from neuronal dynamics is clear, what are the key methodological or performance advantages of framing this problem within the SNN paradigm, as opposed to approaching it from a pure network quantization perspective? This clarification would help position the work more precisely.

---

> ### Author Response · Authors · 2025-11-22
>
> We sincerely thank the reviewer for your time and constructive comments. The response to your questions is given piece by piece as follows.
>
> __Weakness 1:__
>
> > Redefining the SNN Paradigm: By introducing continuous-valued transformations during training, the PIT method blurs the lines between traditional SNNs and ANNs with specialized quantization. This raises the question of whether we are still strictly in the SNN domain or creating a new class of SNN-inspired, highly efficient hybrid networks. The paper could benefit from a more explicit discussion of its philosophical positioning within the broader neural network landscape.
>
> __Reply:__ We thank the reviewer for this insightful comment on the philosophical positioning of our work. We appreciate the opportunity to clarify our perspective on the SNN paradigm and the role of our introduced PIT.
>
> **Our philosophical positioning:**
> This research, by analyzing information propagation from a transformation perspective, provides a more fundamental view of the relationship between ANNs and SNNs, rather than merely finding optimal mathematical mappings between ReLU activation layers and spiking neuron layers. This work offers three key insights for the community:
> 1. To mitigate the information loss caused by the firing operation, we should adjust the modulation mechanism before and after firing based on the membrane potential distribution to ensure effective information propagation pathways.
> 2. Due to the inherent temporal dependency of SNNs, the proposed method should be time-varying and heterogeneous along the spatiotemporal dimension, rather than simply applying spatial designs with temporal sharing, as employed in many prior QNN works.
> 3. We need to redesign continuous relaxations or surrogate gradients to address the discrete, non-differentiable nature of spike generation during training, thereby improving the generalizability of SNNs.
>
>
> **Distinction to established practices:**
> Our design philosophy parallels yet departs from widely accepted practices in the SNN community:
>
> 1. **Batch normalization**: Unlike previous BN methods [1,2] for SNNs that apply to input currents, our PIT reorganizes membrane potentials along the spatiotemporal dimension to mitigate distribution shift and reduce information loss caused by firing operations (Eq. 10-11).
>
> 2. **Surrogate gradients**: While the field has long used pre-defined surrogate functions (e.g., sigmoid, arctangent) during training to address non-differentiable spike firing, our work proposes an auxiliary gradient correction term designed to mitigate gradient mismatch and oscillation phenomena, facilitating rapid convergence and improved generalization (Figure 5).
>
> 3. **ANN-to-SNN conversion**: Unlike conversion methods that establish mathematical mappings between ReLU activation layers and spiking neuron layers, we start directly from SNNs and analyze their representational limitations (theoretical analysis in Sections 3.2 and 3.5). PIT alleviates information loss from firing operations across spatiotemporal dimensions by adapting to evolving membrane potential distributions in a conjugate manner.
>
>
> We agree with the reviewer that this work can be viewed as creating SNN-inspired, highly efficient networks that leverage both biological plausibility (spike-based processing) and engineering pragmatism (learnable transformations). We believe this hybrid approach represents a promising direction for novel SNN design, preserving the advantages of spike-based and event-driven computation.
>
> We have added an additional discussion in the revised manuscript positioning PIT within the broader neural network landscape and emphasizing its role in bridging continuous optimization during training with discrete computation during inference.
>
> [1] Kim, Y., Panda, P. Revisiting batch normalization for training low-latency deep spiking neural networks from scratch. Frontiers in neuroscience, 2021.
>
> [2] Duan, Chaoteng, et al. Temporal effective batch normalization in spiking neural networks. Advances in Neural Information Processing Systems, 2022.

---

> ### Author Response · Authors · 2025-11-22
>
> __Weakness 2:__
>
> > Unaddressed Trade-off between Temporal Dynamics and Bit-Depth. The paper relies on the T×D metric to claim comparable latency with prior work, assuming that trading temporal steps (T) for higher bit-depth (D) is an equitable exchange. This assumption is could be a weakness as it overlooks a critical trade-off. By decomposing a D-bit spike into D sequential sub-steps within a single logical timestep (as detailed in Appendix A), the model's architecture fundamentally shifts from a truly sequential processor to a static one with enhanced precision. A model with high T and low D (e.g., T=4, D=1) can integrate information across distinct and potentially distant time steps, making it suitable for tasks with genuine temporal dependencies. In contrast, the authors' preferred configuration of low T and high D (e.g., T=1, D=4) largely sacrifices this temporal processing capability in favor of higher representational richness within an isolated moment. The paper fails to acknowledge or discuss this fundamental architectural trade-off. This limits the claimed generality of the method, as its effectiveness on truly temporal tasks (like video analysis or complex time-series forecasting) compared to traditional high-T SNNs is not established and remains questionable. Furthermore, the equivalence of latency under the T×D metric is an idealization that may not hold on real hardware, where micro-looping overhead for high D could introduce different latency characteristics compared to processing distinct high-T steps.
>
>
> __Reply:__ We sincerely thank the reviewer for raising this important concern regarding the trade-off between temporal dynamics and bit-depth.
>
> In fact, the trade-off between temporal steps and bit-depth in this work is designed to enable SNNs to adapt to input data patterns and distributions for better temporal processing and modeling capability while maintaining low inference latency.
>
> Our method's design philosophy is oriented toward two scenarios:
> 1. **Static image datasets** (CIFAR10/100, ImageNet): Where temporal structure is artificially introduced through replication, it is reasonable to set high D with low T for better training efficiency (Table 2). Nevertheless, we conducted additional experiments on CIFAR10 to evaluate the impact of different configurations on the model's performance, as reported in Table 8 and Table 9.
>
> 2. **Neuromorphic datasets** (CIFAR10-DVS, DVS-Gesture): Where temporal dependencies exist within event streams, our method achieves sota performance with low D and high T (Tables 7-8). Our setup is consistent with the compared methods, which demonstrates that our approach effectively improves sequential learning capability by excluding the effects introduced by multi-bit spikes (by setting D to 1).
>
> Regarding hardware considerations, we acknowledge that the T×D latency equivalence is an idealization. Real hardware may exhibit different characteristics due to its customized design.
> Recently developed Intel's Loihi 2 neuromorphic chip [3] supports multi-bit firing neurons, which aligns well with our design and facilitates practical deployment. However, we acknowledge that this requires more comprehensive hardware benchmarking across diverse platforms.
>
>
> We have included a detailed discussion on the relationship between the T vs. D trade-off and the dataset, as well as the application scope of this work, in the revised manuscript.
> We thank the reviewer for highlighting this important problem, which will significantly strengthen the manuscript's positioning and clarification about the method's scope.
>
> [3] Orchard, G., et al. Efficient neuromorphic signal processing with Loihi 2. IEEE SiPS, 2021.

---

> > ### Author Response · Authors · 2025-11-22
> >
> > __Q1:__
> >
> > > On the Fairness of the T×D Comparison: The paper uses T×D as a measure of equivalent inference latency, comparing their method (e.g., T=1, D=4) with prior works (e.g., T=4, D=1 or T=2, D=2). While T×D can be a proxy for total operations in some cases, different combinations of T and D can have different implications for actual hardware latency, memory access patterns, and the model's dynamic properties. For instance, a larger T implies longer sequential dependencies, while a larger D might require more complex computation within a single step. Could the authors discuss this trade-off in more detail? What are the limitations of this comparison metric?
> >
> >
> > __Reply:__ We thank the reviewer for this important question regarding the fairness and limitations of the T×D comparison metric. We provide a detailed discussion below.
> >
> > Regarding the rationale for using T×D as a comparison metric, we would like to politely clarify the following reasons:
> >
> > 1. **Following established practice in prior works**: The T×D metric has been adopted in the previous studies [4,5,6] as a measure for comparing inference latency and computational overhead, making it a reasonable basis for fair comparison with existing methods.
> >
> > 2. **Computational overhead approximation**: T×D captures the total number of neuronal update operations required, serving as a tractable approximation of computational cost, which is similar to the bit budget used in quantization literature [7]. This provides a unified framework for comparing different configurations.
> >
> > 3. **Hardware-agnostic comparison metric**: In the absence of standardized neuromorphic hardware benchmarks, T×D provides a hardware-agnostic metric that enables comparison across different platforms and implementations.
> >
> >
> > We acknowledge that T×D is an imperfect proxy with several aspects:
> >
> > 1. **Different computational paradigms**: As the reviewer correctly points out, larger T implies longer sequential dependencies and temporal integration capabilities, while larger D requires more complex computation within individual timesteps. These represent fundamentally different computational paradigms with distinct implications for model dynamics and expressiveness.
> >
> > 2. **Hardware implementation differences**:
> >    - **Asynchronous implementation** (supported by chips like Speck [8]): High-D spiking neuron model can fire D consecutive spikes in a very short time window without requiring a global clock, offering significant advantages in power and latency when D is moderate and firing rates are low. However, extremely large D values may exceed communication bandwidth in scenarios with dense spike firing.
> >
> >    - **Synchronous implementation** (supported by chips like Tianjic [9]): Expands T timesteps to T×D timesteps with a global clock, where latency scales more linearly with the T×D product.
> >
> > 3. **Memory access patterns**: High-T models repeatedly access weights across timesteps (though can exploit temporal sparsity), while high-D models may require different memory access strategies depending on synchronous versus asynchronous implementation.
> >
> >
> >
> > [4] Yao, M., et al. Scaling spike-driven transformer with efficient spike firing approximation training. IEEE TPAMI, 2025.
> >
> > [5] Luo, X., et al. Integer-valued training and spike-driven inference spiking neural network for high-performance and energy-efficient object detection. ECCV 2024.
> >
> > [6] Fan, L., et al. A multisynaptic spiking neuron for simultaneously encoding spatiotemporal dynamics. Nature Communications, 2025.
> >
> > [7] Shen, G., et al. Are conventional snns really efficient? a perspective from network quantization. CVPR, 2024.
> >
> > [8] Yao, M., et al. Spike-based dynamic computing with asynchronous sensing-computing neuromorphic chip. Nature Communications, 2024.
> >
> > [9] Pei, J. et al. Towards artificial general intelligence with hybrid tianjic chip architecture. Nature, 2019

---

> ### Author Response · Authors · 2025-11-22
>
> __Q2:__
>
> > On the Sensitivity of the Rectified Surrogate Gradient: The rectified surrogate gradient proposed in Eq. (13) is an interesting design that aims to reduce training oscillations by penalizing inputs near the rounding boundary. The computation introduces a hyperparameter λ, which is set to 0.01 in Appendix B. How sensitive is the model's performance to the choice of λ? A sensitivity analysis on the value of λ would be valuable for demonstrating the robustness and practical utility of this gradient correction term.
>
> __Reply:__ We thank the reviewer for this question regarding the sensitivity of our rectified surrogate gradient design. We conducted ablation experiments on the CIFAR10 and CIFAR100 datasets using ResNet18 to evaluate the impact of $\lambda$ values. The results are summarized below:
>
> | Dataset  | Architecture | Params | T×D | $\lambda$ | Accuracy |
> | -------- | ------------ | ------ | --- | ------ | -------- |
> | CIFAR10  | ResNet18     | 11.18M | 1×4 | 0.1    | 95.73%   |
> |    | ResNet18     | 11.18M | 1×4 | 0.01   | 95.86%   |
> |    | ResNet18     | 11.18M | 1×4 | 0.001  | 95.70%   |
> | CIFAR100 | ResNet18     | 11.18M | 1×4 | 0.1    | 78.10%   |
> |   | ResNet18     | 11.18M | 1×4 | 0.01   | 78.83%   |
> |   | ResNet18     | 11.18M | 1×4 | 0.001  | 79.01%   |
>
> Our experiments reveal that the model demonstrates stable performance across a reasonable range of $\lambda$ values, from 0.1 to 0.001.
> The relatively wide range ([0.001, 0.01]) demonstrates the practical utility of our gradient correction term, as users do not need extensive hyperparameter tuning to achieve good performance.
> We have included this sensitivity analysis in the revised manuscript to provide readers with practical guidance on hyperparameter selection.
>
> __Q3:__
>
> > On the Distinction from Quantized Neural Networks (QNNs): In the low-latency regime of T=1 and D>1, the proposed model strongly resembles a Quantized Neural Network, where the integer spike value acts as a quantized activation and the neuron's dynamics effectively become a stateful quantization function. Could the authors elaborate on the fundamental distinctions between their approach (specifically in the T=1 case) and mainstream QNN research? While the inspiration from neuronal dynamics is clear, what are the key methodological or performance advantages of framing this problem within the SNN paradigm, as opposed to approaching it from a pure network quantization perspective? This clarification would help position the work more precisely.
>
> __Reply:__ We thank the reviewer for this insightful question regarding the distinction between our approach and mainstream Quantized Neural Networks (QNNs). We provide a detailed discussion below on the key differences and contributions of our work.
>
> **Fundamental distinctions from QNN research:**
>
> 1. **Novel information propagation analysis perspective**: As presented in Section 3.2, our analysis reveals that the information loss caused by firing operations depends on the spatiotemporal distribution of membrane potentials before firing. This motivates our input-stream-aware adaptation mechanism in PIT, which differs fundamentally from QNN approaches that primarily approximate ReLU activation patterns in the spatial dimension alone.
>
> 2. **Spatiotemporal heterogeneity design**: In QNNs, temporal or sequence dependencies are typically absent, leading to the same quantization strategy being shared across temporal dimensions. In contrast, our proposed PIT is time-varying and heterogeneous along the spatiotemporal dimension, motivated by our identification of inherent temporal dependencies in potential updates determined by the decay factor, residual potential, and input stream (Eqs. 10-11). This spatiotemporal adaptation is a distinctive feature not present in conventional QNN frameworks.
>
> 3. **Additional trade-off dimensions**: Mainstream QNN research focuses on the trade-off between weight bits and activation bits to reduce computational overhead. Our work introduces two orthogonal trade-offs:
>
>    - **T vs. D trade-off**: Different combinations of total timesteps T and maximum integer firing value D not only produce distinct firing patterns (Figure 6) but also have different implications for hardware implementation modes (asynchronous vs. synchronous), as illustrated in our response to **Q1**.
>
>    - **Model expressivity vs. computational overhead trade-off**: As discussed in Appendix Section F, the PIT can take forms beyond the diagonal structure. For example, a tri-diagonal PIT form allows neurons to receive modulation signals (sub-threshold potentials) from neighboring neurons before firing, partially mimicking lateral inhibition and excitatory-inhibitory balance mechanisms, a biological principle that has no direct analogue in QNN research.
>
> We have added a discussion in the revised manuscript explicitly contrasting our approach with existing works to clarify our unique contributions and positioning.

---

### Official Review · Reviewer_CsiP · 2025-10-31

**Soundness:** 2
**Presentation:** 2
**Contribution:** 2
**Rating:** 4
**Confidence:** 4

**Summary:**

This paper proposes Parametric Invertible Transformation (PIT), a linear transform integrated conjugately around the firing/quantization step of spiking neurons. The motivation is to enhance the representational capabilities of SNNs. A rectified surrogate gradient is introduced that penalizes proximity to rounding decision boundaries to mitigate oscillations. A theoretical section defines “representation space/capacity” and shows that PIT can expand the space by offering more degrees of freedom while preserving spike-driven inference. Experiments on static and neuromorphic datasets demonstrate that the method improves the performance across ResNet-like SNNs and spike-driven transformers.

**Strengths:**

1. Good empirical performance. The reported multi-dataset and multi-architecture improvements demonstrate substantial gains under various scenarios.
2. Ablation results in Appendix show the importance of each component.
3. Theoretical energy consumption analysis show that PIT yields lower energy through firing-rate reduction.

**Weaknesses:**

1. Expressivity claim is overstated. PIT is largely a reparameterization: the linear transformation is a per-channel rescaling of weights which does not change the function class. It is akin to learnable pre-quantization scaling or BN-like affine parameters rather than a fundamentally more expressive operator. Gains plausibly stem from better quantization alignment and training dynamics, not representational power. This aligns with the paper’s own capacity formula that is identical to the non-PIT case. It largely undercuts the title/abstract’s emphasis on “advancing spatiotemporal representations”.
2. Time-varying linear transformation is not friendly for neuromorphic deployment. The transformation $A_t$ varies over time, meaning that per-t weight variants at inference is required. Standard neuromorphic hardware does not support such kind of operations, and it prevents the model from extending to more time steps.
3. Positioning vs. prior works. PIT’s diagonal scaling and distribution-aware initialization feel close to learnable pre-quantization scaling (similar to Real Spike’s reparameterization). The difference is primarily the conjugate placement and the specific initialization method. A clearer differentiation for contributions is required.

**Questions:**

See Weaknesses.

---

> ### Author Response · Authors · 2025-11-22
>
> Thank you very much for your precious time and recognition of our work. We first list your advice and questions, then give our detailed answers.
>
>
> __Weakness 1:__
>
> > Expressivity claim is overstated. PIT is largely a reparameterization: the linear transformation is a per-channel rescaling of weights which does not change the function class. It is akin to learnable pre-quantization scaling or BN-like affine parameters rather than a fundamentally more expressive operator. Gains plausibly stem from better quantization alignment and training dynamics, not representational power. This aligns with the paper’s own capacity formula that is identical to the non-PIT case. It largely undercuts the title/abstract’s emphasis on “advancing spatiotemporal representations”.
>
> __Reply:__ We sincerely thank the reviewer for this insightful comment. As discussed in Section 3.5, our analysis of representation examines two aspects: **(1) representation space** (the set of representable configurations) and **(2) representation capacity** (the cardinality of that space).
> We acknowledge the reviewer's point that PIT does not change the theoretical capacity (Eq. 19 vs. Eq. 17). However, PIT fundamentally expands the representation space by introducing additional degrees of freedom for spatiotemporal variations. With PIT (Eq. 18), the representation space is extended as: $ \\{ \sum_{i=1}^T{\sum_{j=1}^N{a_{ij}k_js_{ij}}}\mid a_{ij}\in \mathbb{R} ,k_j\in \mathbb{R} ,s_{ij}\in \\{0,\cdots ,D\\} \\}$.
>
> The introduction of time-varying parameters provides several key benefits: (1) expands the accessible representation space across spatial and temporal dimensions, (2) enables input-distribution-aware adaptation (Section 3.3, Eq. 12), and (3) improves generalization ability through rectified gradient flow (Section 3.4, Eq. 13).
>
> Collectively, while PIT maintains theoretical capacity, it fundamentally enhances the network's ability to learn, access, and utilize diverse spatiotemporal patterns through time-varying, conjugate transformations. The gains arise from both improved quantization alignment and expanded practical expressivity of the representation space.
>
>
>
> __Weakness 2:__
>
> > Time-varying linear transformation is not friendly for neuromorphic deployment. The transformation $A_t$ varies over time, meaning that per-t weight variants at inference is required. Standard neuromorphic hardware does not support such kind of operations, and it prevents the model from extending to more time steps.
>
> __Reply:__ We thank the reviewer for raising this important practical concern. We clarify that PIT is compatible with neuromorphic hardware and does not require dynamic weight updates during inference. Below, we detail the deployment strategy of our proposed PIT for the FPGA-based neuromorphic hardware platform.
> As detailed in Appendix A (Eqs. 22-23), PIT parameters $A_t$ are folded into weight matrices through the reparameterization technique and precomputed across the entire time span before deployment. Specifically:
>
> - For each timestep, we precompute folded weights: $\tilde{W}_t = W A_t$ .
> - These folded weights form a static lookup table indexed by timestep.
> - During inference, we simply retrieve $\tilde{W}_t$ at the current time index and perform standard spike-driven AC operations.
>
> This deployment strategy relies on table lookup, which is supported by standard neuromorphic hardware.
>
> Regarding temporal extension, we want to clarify that $A_t$ in PIT is time-indexed but operates on a fixed temporal window during training and is replicated $D$ times during inference (based on the equivalent timestep transformation $\tilde{T} = T \times D$). Specifically:
> - **For fixed-length inputs** (e.g., static images): The same pre-folded weights apply at each timestep, maintaining spike-driven computation.
> - **For variable-length inputs** (e.g., event-based data stream): By setting a higher firing integer $D$ and reducing the $T$ used during training, we could effectively extend the equivalent timesteps used during inference, $\tilde{T} = T \times D$.

---

> ### Author Response · Authors · 2025-11-22
>
> __Weakness 3:__
>
> > Positioning vs. prior works. PIT’s diagonal scaling and distribution-aware initialization feel close to learnable pre-quantization scaling (similar to Real Spike’s reparameterization). The difference is primarily the conjugate placement and the specific initialization method. A clearer differentiation for contributions is required.
>
> __Reply:__ We thank the reviewer for this comment and provide the following clarification on the key differences between our PIT and prior work [1].
>
> From the methodology perspective, the fundamental differences are illustrated as follows:
> 1. **Conjugate transformation vs. post-firing scaling**: Real Spike applies scaling only after firing, whereas PIT operates in a conjugate manner both before ($A_t^{-1}$ reorganizes potential distribution) and after firing ($A_t$ augments spikes), as illustrated in Figure 1 and Eq. 10. Our method seamlessly combines the adaptive threshold mechanism with augmented output representation in a conjugate manner, facilitating stable information propagation by maintaining variance between inputs and outputs.
>
> 2. **Spatiotemporal decoupling vs. spatial-only**: Real Spike uses spatial parameters $a_j$ (constant across time), while PIT introduces time-varying parameters $a_{ij}$ that enable different quantization strategies at different timesteps (Figure 4c demonstrates this temporal adaptation).
>
> 3. **Input-distribution-aware initialization vs. random initialization**: Our 3-sigma initialization (Eq. 12, Figure 2c) dynamically aligns with input distribution, avoiding the quantization range instability and leveraging invaluable quantization levels.
>
> 4. **Rectified surrogate gradient function vs. predefined surrogate gradient function**: Our proposed rectified surrogate gradient function mitigates the gradient mismatch issue and oscillation phenomena, facilitating rapid convergence and improved generalization (Figure 5).
>
>
> Based on theoretical analysis, as shown in Table 1 (Section 3.5), the representation spaces are fundamentally different:
> - Real Spike: $\{\sum_i \sum_j a_j k_j s_{ij}\}$ (spatial only)
> - PIT: $\{\sum_i \sum_j a_{ij} k_j s_{ij}\}$ (spatiotemporal)
>
> This theoretical distinction leads to consistent performance gains:
> - **CIFAR100** (Table 2): PIT (ResNet19) outperforms Real Spike (ResNet20) by 14.99% in terms of accuracy
> - **CIFAR10-DVS** (Table 11): PIT surpasses Real Spike by 5.92% in terms of accuracy
> - **DVS-Gesture** (Table 12): PIT exceeds Real Spike by 2.02% in terms of accuracy
>
> These substantial improvements across diverse datasets demonstrate that PIT's conjugate design, spatiotemporal decoupling, distribution-aware initialization, and rectified gradient flow provide distinct and complementary contributions beyond prior work.
>
> In summary, this paper provides the following key insights for the future SNN research:
> 1. Due to the inherent spatiotemporal dynamics of SNNs, it is necessary to design time-varying methods to better achieve the trade-off between energy cost (associated with firing density) and performance.
> 2. As analyzed in Section 3.2, SNNs require adaptively adjusting firing activity based on the distribution of membrane potentials to reduce information loss and maintain the variance of inputs and outputs at the same magnitude for stable information flow.
> 3. For gradient-based learning, it is essential to design adaptive surrogate gradients tailored for SNNs to find better solutions with improved generalizability.
>
> The authors believe that these advancements and insights could enhance currently developed SNNs to fully leverage their unique spatiotemporal dynamics. We have included additional discussion in the revised manuscript.
>
> [1] Guo, Y., et al. "Real spike: Learning real-valued spikes for spiking neural networks." ECCV, 2022.

---

### Author Response · Authors · 2025-12-01

Dear PC, SAC, AC,

We sincerely appreciate the great effort and time you have spent reviewing each submission. Here, we would like to provide a brief summary of the rebuttal status to assist your decision-making.

|Reviewer|Initial Rating|Status|Note|
|-|-|-|-|
|CsiP|4 [2, 2, 2]| No Reply |All concerns answered|
|p6Qe|4 [3, 3, 3]| No Reply |All concerns answered|
|rRf7|6 [3, 3, 3] | No Reply |All concerns answered|
|KE4i|4 [2, 3, 2]| No Reply |All concerns answered|

---

__Our Requests__

Given the lack of response from all reviewers, we would greatly appreciate it if you could review our responses, as we are confident that all their concerns have been fully addressed with comprehensive experimental evidence and theoretical justifications.

For your convenience, we have provided details about our work and discussions with reviewers below.

---

__Recognition from Reviewers__

- Reviewer ```CsiP```:
   - Good empirical performance.
   - Comprehensive ablation results.
   - Extensive energy consumption analysis.
- Reviewer ```p6Qe```:
   - The novelty of the PIT concept.
   - Strong empirical support.
- Reviewer ```rRf7```:
   - Well-written and clearly organized paper.
   - PIT effectively enhances representations in SNNs.
   - The introduced auxiliary gradient correction term facilitates SNN optimization.
   - Comprehensive comparisons with SOTA models.
- Reviewer ```KE4i```:
   - In terms of quality, the components are principled and practical—easy to implement in standard toolchains, trainable end-to-end, and compatible with event-driven deployment.

---

__Main Contributions of Our Work__

1. **Novel Method for Enhancing Spiking Representations:** We introduce PIT to fundamentally enhance SNNs' spatiotemporal representations in a conjugate and time-varying manner, enabling rapid convergence with superior generalizability.

2. **Theoretical Analysis Framework:** We provide a theoretical framework to quantitatively analyze SNNs' representation space and capacity.

3. **State-of-the-Art Performance:** Our method achieves SOTA results on static and neuromorphic datasets. Notably, SEW ResNet-34 with PIT surpasses baselines after training **one epoch** and improves accuracy by **5.62%** on ImageNet-1k, approaching the performance of its full-precision ANN counterpart.

---

__Our Responses to Main Concerns of Each Reviewer__

- Reviewer ```CsiP```
   - **Expressivity of PIT:** We clarified that our proposed PIT expands the representation space by introducing additional degrees of freedom for spatiotemporal variations, as analyzed in Section 3.5.
   - **Neuromorphic Deployment:** We provided explanations of hardware implementation schemes for our PIT and demonstrated that it is supported by neuromorphic hardware.
   - **Comparison with Prior Works:** The fundamental differences between our work and prior works have been illustrated from three aspects: conjugate transformation, spatiotemporal decoupling, and rectified gradient flow.

-  Reviewer ```p6Qe```
   - **Discussion of Philosophical Positioning:** We clarified that our work provides a unique theoretical analysis perspective identifying current bottlenecks, and demonstrates the necessity for time-varying strategies and effective surrogate gradients to improve adaptivity and generalizability.
   - **Trade-off Between Temporal Dynamics and Bit-Depth:** We elaborated on the trade-off between T and D from different application scenarios and hardware implementation.
   - **Sensitivity of the Hyperparameter:** We added an ablation study demonstrating the sensitivity to hyperparameter λ in the surrogate gradient function.

- Reviewer ```rRf7```
   - **Analysis on Memory Cost:** we added memory footprint comparisons between our model and prior works to provide a comprehensive assessment.
   - **Comparison with E-SpikeFormer:** we conducted experiments on ImageNet-1k using E-SpikeFormer architectures of different scales for comparison.

- Reviewer ```KE4i```
   - **The Difference from QAT:** We comprehensively distinguished our work from QAT through: (1) time-varying dynamics with recurrent dependencies, and (2) the trade-off between firing sparsity and performance.
   - **Add QAT Baselines:** We conducted experiments on static and neuromorphic datasets, demonstrating consistently superior performance obtained by our method.
   - **Temporal Dynamics Analysis:** We conducted additional ablation studies on CIFAR-10 and DVS-Gesture to investigate the impact of different T×D combinations on performance.
   - **Energy Evidence:** We adopted S-ACE and NS-ACE metrics for energy comparison, demonstrating that our method achieves better accuracy under matched S-ACE budgets and lower NS-ACE.

---

We believe our comprehensive responses with extensive additional experiments fully address all reviewer concerns. We respectfully request favorable consideration from the AC.

Thank you for your time and consideration.

Sincerely,

Authors of Submission 18012

---

### Meta-Review · Area_Chair_Ygro · 2026-01-02

**Summary:**

Here are main concerns of the reviewers:
* Insuffiicient comparison with other approaches and embedding into the literature, which almost all reviewer mentioned
* concerns about the chosen SNN model, whether it is really an SNN or more an ANN, and about the implementation
* concerns about effectiveness of Parametric Invertible Transformation (PIT), also in a sense whether it is based on the correct assumptions

**Reviewer Concerns:**

I believe the authors did address the concerns.

**Reviewer Scores:**

It’s really hard to say how any reviewer would have changed their score if they had taken part more fully in the discussion. Without hearing it from them directly, anything we write here would just be guesswork.

For this paper, the scores were 4,4,6,4. If I need to guess, I would assume that the reviewers would raise their grade by about one or two, which makes it a borderline case. In favor of the authors due to the difficult situation (they also did not get any reply from the reviewers during the rebuttal phase), I would vote for accept.

---

### Decision · Program_Chairs · 2026-01-26

Accept (Poster)